# MPhil: Multi-Prototype Hyperspherical Invariant Learning for Graph Out-of-Distribution Generalization

## Abstract

Out-of-distribution (OOD) generalization has emerged as a critical challenge in graph learning, as real-world graph data often exhibit diverse and shifting environments that traditional models fail to generalize across. A promising solution to address this issue is graph invariant learning (GIL), which aims to learn invariant representations by disentangling label-correlated invariant subgraphs from environment-specific subgraphs. However, existing GIL methods face two major challenges: (1) the difficulty of **capturing and modeling diverse environments** in graph data, and (2) the **semantic cliff**, where invariant subgraphs from different classes are difficult to distinguish, leading to poor class separability and increased misclassifications. To tackle these challenges, we propose a novel method termed **M**ulti-**P**rototype **H**yperspherical **I**nvariant **L**earning (MPhil), which introduces two key innovations: (1) *invariant learning in hyperspherical space*, enabling robust invariant feature extraction and prototypical learning in a highly discriminative space, and (2) *class prototypes as intermediate variables*, which eliminate the need for explicit environment modeling in GIL and mitigate the semantic cliff issue through multi-prototype-based classification. Derived from the theoretical framework of GIL, we introduce two novel objective functions: the *invariant prototype matching loss* to ensure samples are matched to the correct class prototypes, and the *prototype separation loss* to increase the distinction between prototypes of different classes in the hyperspherical space. Extensive experiments on 11 OOD generalization benchmark datasets demonstrate that MPhil achieves state-of-the-art performance, significantly outperforming existing methods across graph data from various domains and with different distribution shifts. The source code of MPhil is available at https://anonymous.4open.science/r/MPHIL-23C0/.

## 1 Introduction

Graph Neural Networks (GNNs) have made remarkable advancements in modeling and learning from graph-structured data across various scenarios, including, but not limited to, social networks (Fan et al., 2020; Chang et al., 2021), molecules (Shui & Karypis, 2020; Liu et al., 2023), and knowledge graphs (Zhang et al., 2020; Ji et al., 2021). Despite the powerful representational capabilities of GNNs, their success often relies on the assumption that the training and testing data follow the same distribution. Unfortunately, such an assumption rarely holds in most real-world applications, where out-of-distribution (OOD) data from different distributions often occurs (Liu et al., 2021). Empirical evidence has shown that GNNs often struggle to maintain performance when tested on OOD data that differ significantly from the training set (Wang et al., 2024; Li et al., 2022a). These vulnerabilities underscore the critical need to enhance the OOD generalization capabilities of GNNs, which has become a rapidly growing area of research (Gui et al., 2022; Ji et al., 2023).

Building on the success of invariant learning in addressing OOD generalization challenges in image data (Creager et al., 2021; Ye et al., 2022), graph invariant learning (GIL) has recently emerged as a prominent solution for tackling its counterpart problem in graph data (Li et al., 2022c; Fan et al., 2022; Miao et al., 2022; Wu et al., 2022b). The fundamental assumption underlying GIL is that each graph sample can be divided into two distinct components, namely *invariant subgraph* and *environment subgraph*. The former exhibits deterministic and solid predictive relationships with the label of the

graph sample, while the latter may show spurious correlations with the labels and can vary significantly in response to distribution shifts (Yang et al., 2022; Chen et al., 2023). By effectively separating invariant and environment information, GIL-based approaches can learn invariant representations from the former and make reliable predictions. To achieve accurate separation, existing methods primarily focus on capturing and modeling the environmental subgraphs, employing carefully designed loss functions to minimize the correlations between the predicted environmental subgraphs and the labels (Gui et al., 2023; Piao et al., 2024), or utilizing data augmentation strategies to simulate potential environments in wild data (Wu et al., 2022a; Jia et al., 2024).

Although GIL-based methods are theoretically viable, they underestimate the **difficulties in capturing environment information** in real-world graph data. These difficulties can be attributed to the *diversity*, *distinguishability*, and *lack of labels* of practical graph environments. Specifically, the environments of graph data can exhibit substantial diversity in terms of sizes, shapes, and topological properties. Taking molecular graphs as an example (Fig. 1(a)), the invariant subgraph can be a specific functional group, while the environmental subgraphs may display varied patterns, such as different scaffolds, side chains, or bonding configurations that modify the overall structure (Zhuang et al., 2023). In this context, even with augmented or reorganized environments during training, GNN models struggle to identify all forms of environment subgraphs in real-world OOD samples. Moreover, unlike image data where environments can be clearly segmented at the pixel level, the structural boundaries between invariant and environmental subgraphs in graph data are often ambiguous (see the examples in Fig. 1(a)), which leads to poor distinguishability of environments. Moreover, unlike certain visual datasets (Lin et al., 2022) where environment labels (such as background or image style) are available for model training, the environments in graph data are highly complex, making it difficult to obtain corresponding labels to aid in capturing environmental information. Given the above difficulties, most existing GIL-based approaches demonstrate suboptimal performance in identifying environmental information, particularly in complex graph data, as empirically demonstrated by (Chen et al., 2023). Consequently, a natural question arises: *(RQ1) Can we consider a more feasible way to learn invariant representations on graphs without explicitly modeling the environment information?*

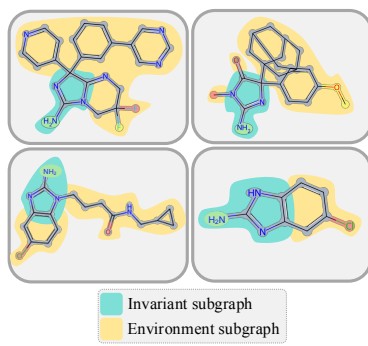

(a) Diversity of environments.

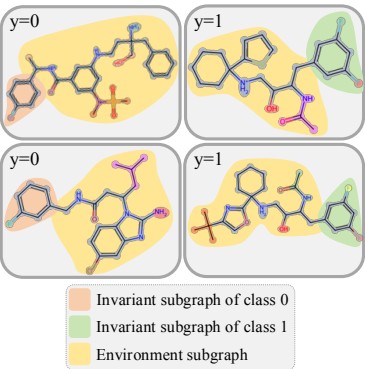

(b) Semantic cliff across classes.

Figure 1: Case examples.

In addition to the challenges posed by environment capturing, another critical obstacle, referred to as the **semantic cliff across different classes** (Xia et al., 2023), also hinders current GIL-based approaches from making reliable decisions on OOD graph data. To be more specific, the semantic cliff issue refers to a situation where invariant subgraphs or representations from different classes share significant similarities, making them difficult to distinguish from one another (Van Tilborg et al., 2022). For instance, as shown in Fig. 1(b), invariant subgraphs (e.g., functional groups) across different molecular classes often share similar structural characteristics, with distinctions frequently limited to a single atom or bond. In such cases, the decision boundaries between invariant representations can become blurred, exacerbating the challenge of separating classes, particularly when the invariant and environment subgraphs are not distinctly identifiable. Nevertheless, the existing OOD generalization GNNs overemphasize the extraction of invariant information while neglecting the distinction between invariant subgraphs belonging to different classes, which may lead to their sub-optimal performance. Given this shortage, a natural follow-up question arises: *(RQ2) How can we develop a more robust OOD generalization approach that better discriminates between invariant representations across different classes?*

To answer the above research questions, in this paper, we propose a novel **M**ulti-**P**rototype **H**yperspherical **I**nvariant **L**earning (MPHIL for short) method. Building upon and enhancing the theoretical framework of GIL, MPHIL introduces two advanced features: 1) **invariant learning in**

**hyperspherical space**, which ensures the separability and informativeness of the learned invariant representations, and 2) **class prototype as intermediate variable**, which eliminates the need for explicit environment modeling in invariant learning and enables flexible decision boundaries for prediction. More specifically, we derive a more practical invariant learning objective based on prototypical learning in hyperspherical space, incorporating two well-crafted loss terms. To address *(RQ1)*, we design an *invariant prototype matching loss* ($\mathcal{L}_{IPM}$) that ensures samples from the same class are assigned to the same class prototype in hyperspherical space. In this way, $\mathcal{L}_{IPM}$ can allow the model to extract robust invariant features across varying environments without explicitly modeling them. To answer *(RQ2)*, we produce a *prototype separation loss* ($\mathcal{L}_{PS}$) that pulls the prototypes belonging to the same class closer together while ensuring those from different classes remain dissimilar. In this way, $\mathcal{L}_{PS}$ enhances the class separability in the context of OOD generalization and mitigates the semantic cliff issue in graph data. Innovatively, we propose to assign multiple prototypes for each class, which ensures adaptable decision boundaries and greater tolerance to environmental changes. To sum up, the main contributions of this paper are as follows:

- **Framework.** Derived from the objective of GIL, we introduce a novel invariant learning framework that leverages hyperspherical space and prototypical learning, ensuring robust invariant representation learning while reducing the need for explicit graph environment modeling.

- **Methodology.** Based on the new GIL framework, we develop a new graph OOD generalization method termed MPHIL. MPHIL incorporates two effective loss terms to enhance intra-class invariance and inter-class separability, with a multi-prototype mechanism to handle diverse environments.

- **Experiments.** We conduct extensive experiments to validate the effectiveness of MPHIL, and the results demonstrate its superior generalization ability compared to state-of-the-art methods across various types of distribution shifts.

## 2 PRELIMINARIES AND BACKGROUND

In this section, we introduce the preliminaries and background of this work, including the formulation of the graph OOD generalization problem, graph invariant learning, and hyperspherical embeddings. A more comprehensive literature review can be found in Appendix A.

**Problem Formulation.** In this paper, we focus on the OOD generalization problem on graph classification tasks (Li et al., 2022b; Jia et al., 2024; Fan et al., 2022; Wu et al., 2022b). We denote a graph data sample as $(G, y)$, where $G \in \mathcal{G}$ represents a graph instance and $y \in \mathcal{Y}$ represents its label. The dataset collected from a set of environments $\mathcal{E}$ is denoted as $\mathcal{D} = \{\mathcal{D}^e\}_{e \in \mathcal{E}}$, where $\mathcal{D}^e = \{(G_i^e, y_i^e)\}_{i=1}^{n^e}$ represents the data from environment $e$, and $n^e$ is the number of instances in environment $e$. Each pair $(G_i^e, y_i^e)$ is sampled independently from the joint distribution $P_e(\mathcal{G}, \mathcal{Y}) = P(\mathcal{G}, \mathcal{Y}|e)$. In the context of graph OOD generalization, the difficulty arises from the discrepancy between the training data distribution $P_{e_{tr}}(\mathcal{G}, \mathcal{Y})$ from environments $e_{tr} \in \mathcal{E}_{tr}$, and the testing data distribution $P_{e_{te}}(\mathcal{G}, \mathcal{Y})$ from unseen environments $e_{te} \in \mathcal{E}_{test}$, where $\mathcal{E}_{te} \neq \mathcal{E}_{train}$. The goal of OOD generalization is to learn an optimal predictor $f : \mathcal{G} \to \mathcal{Y}$ that performs well across both training and unseen environments, $\mathcal{E}_{all} = \mathcal{E}_{tr} \cup \mathcal{E}_{te}$, i.e.,

$$\min_{f \in \mathcal{F}} \max_{e \in \mathcal{E}_{\text{all}}} \mathbb{E}_{(G^e, y^e) \sim P_e}[\ell(f(G^e), y^e)], \tag{1}$$

where $\mathcal{F}$ denotes the hypothesis space, and $\ell(\cdot, \cdot)$ represents the empirical risk function.

**Graph Invariant Learning (GIL).** Invariant learning focuses on capturing representations that preserve consistency across different environments, ensuring that the learned invariant representation $\mathbf{z}_{inv}$ maintains consistency with the label $y$ (Mitrovic et al., 2020; Wu et al., 2022b; Chen et al., 2022b). Specifically, for graph OOD generalization, the objective of GIL is to learn an invariant GNN $f := f_c \circ g$, where $g : \mathcal{G} \to \mathcal{Z}_{inv}$ is an encoder that extracts the invariant representation from the input graph $G$, and $f_c : \mathcal{Z}_{inv} \to \mathcal{Y}$ is a classifier that predicts the label $y$ based on $\mathbf{z}_{inv}$. From this perspective, the optimization objective of OOD generalization, as stated in Eq. (1), can be reformulated as:

$$\max_{f_c, g} I(\mathbf{z}_{inv}; y), \text{ s.t. } \mathbf{z}_{inv} \perp e, \forall e \in \mathcal{E}_{tr}, \mathbf{z}_{inv} = g(G), \tag{2}$$

where $I(\mathbf{z}_{inv}; y)$ denotes the mutual information between the invariant representation $\mathbf{z}_{inv}$ and the label $y$. This objective ensures that $\mathbf{z}_{inv}$ is independent of the environment $e$, focusing solely on the most relevant information for predicting $y$.

**Hyperspherical Learning.** Hyperspherical learning enhances the discriminative ability and generalization of deep learning models by mapping feature vectors onto a unit sphere (Liu et al., 2017). To learn a hyperspherical embedding for an input graph $G$, the representation vector $\mathbf{z}$ is mapped into hyperspherical space with arbitrary linear or non-linear projection functions, followed by normalization to ensure that the projected vector $\hat{\mathbf{z}}$ lies on the unit hypersphere. To make classification prediction, the hyperspherical embeddings $\hat{\mathbf{z}}$ are modeled using the von Mises-Fisher (vMF) distribution (Ming et al., 2022), with the probability density for a unit vector in class $c$ is given by:

$$p(\hat{\mathbf{z}}; \boldsymbol{\mu}^{(c)}, \kappa) = Z(\kappa) \exp(\kappa \boldsymbol{\mu}^{(c)\top} \hat{\mathbf{z}}), \tag{3}$$

where $\boldsymbol{\mu}_c$ denotes the prototype vector of class $c$ with the unit norm, serving as the mean direction for class $c$, while $\kappa$ controls the concentration of samples around $\boldsymbol{\mu}_c$. The term $Z(\kappa)$ serves as the normalization factor for the distribution. Given the probability model in Eq.(3), the hyperspherical embedding $\hat{\mathbf{z}}$ is assigned to class $c$ with the following probability:

$$\mathbb{P}\left(y = c \mid \hat{\mathbf{z}}; \{\kappa, \boldsymbol{\mu}^{(i)}\}_{i=1}^{C}\right) = \frac{Z(\kappa) \exp\left(\kappa \boldsymbol{\mu}^{(c)\top} \hat{\mathbf{z}}\right)}{\sum_{i=1}^{C} Z(\kappa) \exp\left(\kappa \boldsymbol{\mu}^{(i)\top} \hat{\mathbf{z}}\right)} = \frac{\exp\left(\boldsymbol{\mu}^{(c)\top} \hat{\mathbf{z}}/\tau\right)}{\sum_{i=1}^{C} \exp\left(\boldsymbol{\mu}^{(i)\top} \hat{\mathbf{z}}/\tau\right)}, \tag{4}$$

where $\tau = 1/\kappa$ is a temperature parameter. In this way, the classification problem is transferred to the distance measurement between the graph embedding and the prototype of each class in hyperspherical space, where the class prototype is usually defined as the embedding centroid of each class.

## 3 METHODOLOGY

In this section, we present the proposed method, **M**ulti-**P**rototype **H**yperspherical **I**nvariant **L**earning (MPHIL). In Sec. 3.1, we first derive our general framework based on the learning objective graph invariant learning (GIL). Then, we describe the specific designs of the components in MPHIL, including hyperspherical invariant representation learning (Sec. 3.2), multi-prototype classifier (Sec. 3.3), and learning objectives (Sec. 3.4). The overall learning pipeline of MPHIL is shown in Fig. 2.

### 3.1 PROTOTYPICAL HYPERSPHERICAL INVARIANT LEARNING FRAMEWORK

The objective of GIL (i.e., Eq. (2)) aims to maximize the mutual information between the invariant representation $\mathbf{z}_{inv}$ and the label $y$, while ensuring that $\mathbf{z}_{inv}$ remains independent of the environment $e$. However, directly optimizing this objective with such strict constraints is challenging due to the difficulty of modeling environments in graph data. To make the optimization more tractable, we relax the independence constraint and introduce a soft-constrained formulation:

$$\min_{f_c, g} -I(y; \mathbf{z}_{inv}) + \beta I(\mathbf{z}_{inv}; e), \tag{5}$$

where $e$ represents the environment to which the current graph belongs, but it cannot be directly observed or accessed. The parameter $\beta$ controls the trade-off between the predictive power of $\mathbf{z}_{inv}$ and its independence from the environment $e$.

Although the relaxed objective Eq. (5) is more feasible, the intractable properties of real-world graph data (i.e., *complex environment information* and *inter-class semantic cliff* as discussed in Sec. 1) still hinder us from learning reliable invariant representations and making accurate predictions with this objective. Specifically, the diversity and complexity of environments make it challenging to explicitly model $e$, leading to the difficulties of minimizing $I(\mathbf{z}_{inv}; e)$. On the other hand, the semantic cliff issue may cause indistinguishable $\mathbf{z}_{inv}$ of samples belonging to different classes, resulting in the hardness of maximizing $I(y; \mathbf{z}_{inv})$ using a simple cross-entropy loss.

To deal with the above challenges, we propose a new GIL framework based on hyperspherical space with prototypical learning. Concretely, we model the invariant representations in a *hyperspherical space* rather than an arbitrary Euclidean space, which enhances the discriminative ability of the learned representations (Mettes et al., 2019). The desirable properties of hyperspherical space allow us to introduce an intermediate variable, the *class prototype* $\boldsymbol{\mu}$, as a bridge between the hyperspherical invariant representation $\mathbf{z}_{inv}$ and label $y$, which alleviate the issues. To be more specific, the class prototypes $\boldsymbol{\mu}$ can directly capture the invariant patterns of each class, which enables the model to

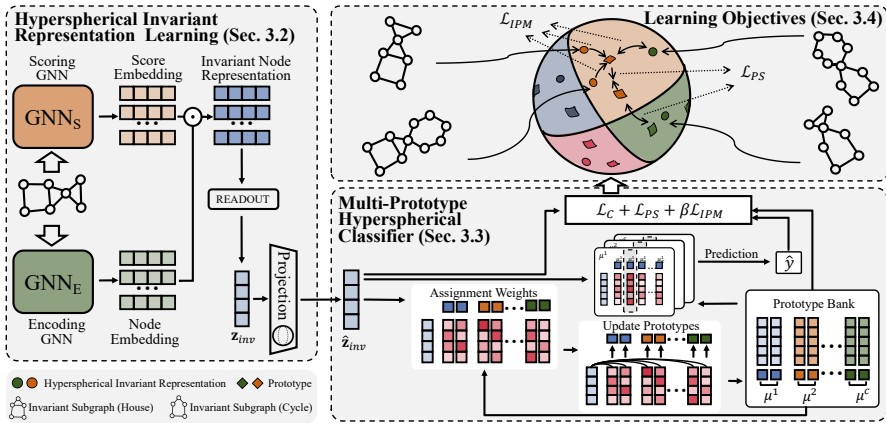

Figure 2: The overall framework of MPHIL. First, a GNN-based model generates the invariant representation and maps it into the hyperspherical space. Then, the classifier makes the prediction based on multiple prototypes. The overall method is trained by a three-term joint objective.

learn reliable invariant representations in the hyperspherical space without explicitly modeling the environment $e$. Moreover, the prototype-based classifier is more robust against the semantic cliff issue, since the semantic gaps between classes can be precisely represented by the distances between prototypes in the hyperspherical space. Formally, the reformed learning objective is as follows, with detailed deductions from Eq. (5) to Eq. (6) provided in Appendix B.1:

$$\min_{f_c, g} \underbrace{-I(y; \hat{\mathbf{z}}_{inv}, \boldsymbol{\mu}^{(y)})}_{\mathcal{L}_{\mathrm{C}}} \underbrace{-I(y; \boldsymbol{\mu}^{(y)})}_{\mathcal{L}_{\mathrm{PS}}} \underbrace{-\beta I(\hat{\mathbf{z}}_{inv}; \boldsymbol{\mu}^{(y)})}_{\mathcal{L}_{\mathrm{IPM}}}, \tag{6}$$

where $\hat{\mathbf{z}}_{inv}$ represents the invariant representation in the hyperspherical space and $\boldsymbol{\mu}^{(y)}$ is the prototype corresponding to class $y$. In the following subsections, we will introduce MPHIL as a practical implementation of the above framework, including the encoder $f_c$ for representation learning (Sec. 3.2), the multi-prototype classifier $g$ (Sec. 3.3), and the three learning objective terms (Sec. 3.4).

## 3.2 HYPERSPHERICAL INVARIANT REPRESENTATION LEARNING

**Encoder.** In GIL, the goal of the encoder $f$ is to extract invariant representations that are highly correlated with the invariant subgraph of each sample. Nevertheless, explicitly identifying the subgraphs via modeling the selecting probabilities of each node and edge may lead to increased overhead and require more complex network architectures (Zhuang et al., 2023). To mitigate these costs, we adopt a lightweight GNN-based model for efficient invariant representation learning. To be specific, our model includes two GNNs: $\mathrm{GNN}_E$ to encode the input graph $G$ into the latent space, producing the node representation $\mathbf{H}$, and $\mathrm{GNN}_S$ to compute the separation score $S$ for the invariant features:

$$\mathbf{H} = \mathrm{GNN}_E(G) \in \mathbb{R}^{|\mathcal{V}| \times d}, \mathbf{S} = \sigma(\mathrm{GNN}_S(G)) \in \mathbb{R}^{|\mathcal{V}| \times d}, \tag{7}$$

where $|\mathcal{V}|$ is the number of nodes in the graph $G$, $d$ is the latent dimension, and $\sigma(\cdot)$ is the Sigmoid function to constrain $\mathbf{S}$ falls into the range of $(0, 1)$. Then, the invariant representation $\mathbf{z}_{inv}$ is obtained through the following operation:

$$\mathbf{z}_{inv} = \mathrm{READOUT}(\mathbf{H} \odot \mathbf{S}) \in \mathbb{R}^d, \tag{8}$$

where $\odot$ is the element-wise product and $\mathrm{READOUT}(\cdot)$ is an aggregation function (e.g., mean) to generate a graph-level representation.

**Hyperspherical Projection.** After obtaining $\mathbf{z}_{inv}$, the next step is to project it into hyperspherical space. Concretely, the hyperspherical invariant representation $\hat{\mathbf{z}}_{inv}$ can be calculated by:

$$\tilde{\mathbf{z}}_{inv} = \mathrm{Proj}(\mathbf{z}_{inv}), \hat{\mathbf{z}}_{inv} = \tilde{\mathbf{z}}_{inv}/\|\tilde{\mathbf{z}}_{inv}\|_2, \tag{9}$$

where $\mathrm{Proj}(\cdot)$ is an MLP-based projector that maps the representation into another space, and dividing $\tilde{\mathbf{z}}_{inv}$ by its norm constrains the representation vector to unit length. The hyperspherical

projection allows invariant learning to occur in a more discriminative space. More importantly, the hyperspherical space provides a foundation for prototypical learning, enabling the extraction of invariant patterns without modeling environments and addressing the semantic cliff issue.

## 3.3 MULTI-PROTOTYPE HYPERSPHERICAL CLASSIFIER

Following hyperspherical projection, the next step is to construct a prototype-based classifier within the hyperspherical space. In conventional hyperspherical learning approaches (Ke et al., 2022), each class is typically assigned a single prototype. Although this is a straightforward solution, its modeling capabilities regarding decision boundaries are limited, as it may not adequately capture the complexity of the data distribution. More specifically, a single prototype often overfits easy-to-classify samples while failing to consider the harder samples. To address this limitation, we propose a multi-prototype hyperspherical classifier in which *each class is represented by multiple prototypes*. This multi-prototype approach ensures that the classification decision space is more flexible and comprehensive, enabling better modeling of the semantic differences among classes. In the following paragraphs, we will explain how to initialize, update, and use the prototypes for prediction.

**Prototype Initialization.** For each class $c \in \{1, \cdots, C\}$, we assign $K$ prototypes for it, and they can be denoted by $\mathbf{M}^{(c)} \in \mathbb{R}^{K \times d} = \{\boldsymbol{\mu}_k^{(c)}\}_{k=1}^K$. At the beginning of model training, we initialize each of them by $\boldsymbol{\mu}_k^{(c)} \sim \mathcal{N}(\mathbf{0}, \mathbf{I})$, where $\mathcal{N}(\mathbf{0}, \mathbf{I})$ represents a standard multivariate Gaussian distribution. Random initialization can help prevent the issue of mode collapse.

**Prototype Updating.** To ensure that the prototypes can represent the majority of samples in their corresponding classes while preserving stability, we adopt the exponential moving average (EMA) technique to update the prototypes asynchronously according to invariant representation $\hat{\mathbf{z}}_{inv}$. Specifically, the update rule for a batch of $B$ samples is given by:

$$\boldsymbol{\mu}_k^{(c)} := \text{Normalize} \left( \alpha \boldsymbol{\mu}_k^{(c)} + (1 - \alpha) \sum_{i=1}^B \mathbb{1}(y_i = c) \boldsymbol{W}_{i,k}^{(c)} \hat{\boldsymbol{Z}}_{inv,i} \right), \quad (10)$$

where $\alpha$ is the EMA update rate, $\boldsymbol{W}_{i,k}^{(c)}$ is the weight of the $i$-th sample for prototype $k$ in class $c$, $\hat{\boldsymbol{Z}}_{inv,i}$ is the representations of the $i$-th sample, and $\mathbb{1}(y_i = c)$ is an indicator function that ensures the update applies only to samples of class $c$. After each update, the prototype is normalized to maintain its unit norm, ensuring it remains on the hypersphere and the distance calculations take place in the same unit space as $\hat{\boldsymbol{Z}}_{inv,i}$. Each representation $\hat{\boldsymbol{Z}}_{inv,i}$ is associated with multiple prototypes, weighted by the assignment weight vector $\boldsymbol{W}_i^{(c)} \in \mathbb{R}^K$.

**Assignment Weight Calculation.** To ensure that each sample is matched with the most relevant prototype, we introduce an attention-based matching mechanism. This approach computes the attention score between each sample and its class prototypes to determine the assignment weights:

$$\mathbf{Q} = \hat{\boldsymbol{Z}}_{inv,i} \mathbf{W}_Q, K = \mathbf{M}^{(c)} \mathbf{W}_K, \boldsymbol{W}_i^{(c)} = \text{softmax}(\frac{\mathbf{Q}\mathbf{K}^\top}{\sqrt{d'}}), \quad (11)$$

where $\mathbf{W}_Q, \mathbf{W}_K \in \mathbb{R}^{d \times d'}$ are the learnable weight matrices for the samples and the prototypes, respectively, and $d'$ is the dimension of the projected space. The attention mechanism ensures that the prototype $\boldsymbol{\mu}^{(c)}$ most similar to the current $\hat{\boldsymbol{Z}}_{inv,i}$ receives the highest weight, which improves classification accuracy and helps the prototype remain aligned with its class center. In practice, to ensure that each sample can only concentrate on a limited number of prototypes, we further introduce a **top-$n$ pruning** strategy: we only preserve $\boldsymbol{W}_{i,k}^{(c)}$ with the top-$n$ largest values while setting the rest to be 0. The detailed algorithmic process and discussions are provided in Appendix C.2.

**Prototype-Based Prediction.** To make classification decisions with the multi-prototype classifier, we can calculate the prediction probability with the similarity between the invariant representation $\hat{\mathbf{z}}_{inv}$ and the set of prototypes $\boldsymbol{\mu}^{(c)}$ associated with each class, which is defined as:

$$p(y = c | \hat{\mathbf{z}}_{inv}; \{\mathbf{w}^{(c)}, \boldsymbol{\mu}^{(c)}\}_{c=1}^{(C)}) = \frac{\max\limits_{k=1,\ldots,K} w_k^{(c)} \exp\left( \boldsymbol{\mu}_k^{(c)\top} \hat{\mathbf{z}}_{inv}/\tau \right)}{\sum_{j=1}^C \max\limits_{k'=1,\ldots,K} w_{k'}^{(j)} \exp\left( \boldsymbol{\mu}_{k'}^{(j)\top} \hat{\mathbf{z}}_{inv}/\tau \right)}, \quad (12)$$

where $w_k^c$ represents the weight of the $k$-th prototype $\boldsymbol{\mu}_k^{(c)}$ assigned to the current sample for class $c$. After that, the class prediction can be directly obtained by an $\arg\max$ operation.

### 3.4 MPHIL LEARNING OBJECTIVES

In this subsection, we formulate the learning objective terms of MPHIL in Eq. (6), including the invariant prototype matching loss $\mathcal{L}_{\text{IPM}}$, prototype separation loss $\mathcal{L}_{\text{PS}}$, and the classification loss $\mathcal{L}_{\text{C}}$. For $\mathcal{L}_{\text{IPM}}$ and $\mathcal{L}_{\text{PS}}$, we formulate them with contrastive learning loss, which is proved to be an effective mutual information estimator (Sordoni et al., 2021; Xie et al., 2022; Sun et al., 2024). For the term of $-I(y; \hat{\mathbf{z}}_{inv}, \boldsymbol{\mu}^{(y)})$, we show in the Appendix B.2 that it can be implemented with classification loss.

**Invariant Prototype Matching Loss** $\mathcal{L}_{\text{IPM}}$. The challenge of disentangling invariant features from environmental variations lies at the heart of OOD generalization. In our formulation, the misalignment of a sample with an incorrect prototype can be seen as a signal of environmental interference. In contrast, successful alignment with the correct prototype reflects the capture of stable and invariant features. Motivated by this, we design $\mathcal{L}_{\text{IPM}}$ that operates by reinforcing the proximity of samples to their invariant representations and penalizing the influence of environmental factors, implicitly captured through incorrect prototype associations. The loss function is expressed as follows:

$$\mathcal{L}_{\text{IPM}} = -\frac{1}{B} \sum_{i=1}^{B} \log \frac{\sum_{c=y_i} \exp\left(\hat{\mathbf{z}}_i^\top \boldsymbol{\mu}^{(c)}/\tau\right)}{\sum_{c=y_i} \exp\left(\hat{\mathbf{z}}_i^\top \boldsymbol{\mu}^{(c)}/\tau\right) + \sum_{\hat{c} \neq y_i} \exp\left(\hat{\mathbf{z}}_i^\top \boldsymbol{\mu}^{(\hat{c})}/\tau\right)}, \tag{13}$$

where $B$ represents the batch size, with $i$ indexing each sample in the batch. $\hat{\mathbf{z}}_i$ represents the hyperspherical invariant representation, $\boldsymbol{\mu}^{(c)}$ is the correct class prototype, $\boldsymbol{\mu}^{(\hat{c})}$ denotes the prototypes of the incorrect classes $\hat{c} \neq y_i$, and $\tau$ is a temperature factor. This formulation reflects the dual objective of pulling samples towards their class-invariant prototypes while ensuring that the influence of prototypes associated with environmental shifts is minimized. The numerator reinforces the similarity between the sample's invariant representation and its correct prototype, while the denominator introduces competition between correct and incorrect prototypes, implicitly modeling the influence of environmental noise.

**Prototype Separation Loss** $\mathcal{L}_{\text{PS}}$. In hyperspherical space, all invariant $\hat{\mathbf{z}}$ representations are compactly clustered around their respective class prototypes. To ensure inter-class separability, prototypes of different classes must be distinguishable. The prototype separation loss $\mathcal{L}_{\text{PS}}$ is designed to enforce this by maximizing the separation between prototypes of different classes while encouraging the similarity of prototypes within the same class. The loss function is defined as:

$$\mathcal{L}_{\text{PS}} = -\frac{1}{CK} \sum_{c=1}^{C} \sum_{k=1}^{K} \log \frac{\sum_{i=1}^{K} \mathbb{I}(i \neq k) \exp\left((\boldsymbol{\mu}_k^{(c)})^\top \boldsymbol{\mu}_i^{(c)}/\tau\right)}{\sum_{c'=1}^{C} \sum_{j=1}^{K} \mathbb{I}(c' \neq c) \exp\left((\boldsymbol{\mu}_k^{(c)})^\top \boldsymbol{\mu}_j^{(c')}/\tau\right)}, \tag{14}$$

where $C$ represents the total number of classes, $K$ denotes the number of prototypes assigned to each class, $\boldsymbol{\mu}_k^{(c)}$ and $\boldsymbol{\mu}_i^{(c)}$ correspond to different prototypes within the same class $c$, $\boldsymbol{\mu}_k^{(c)}$ and $\boldsymbol{\mu}_j^{(c')}$ represent prototypes from different classes, and $\boldsymbol{\mu}_k^{(c)}$ and $\boldsymbol{\mu}_j^{(c')}$ represent prototypes from different classes. Such an indicator function ensures that the comparisons are made between distinct prototypes, enhancing intra-class similarity and inter-class separation.

**Classification Loss** $\mathcal{L}_{\text{C}}$. To calculate the classification loss with the multi-prototype classifier, we update the classification probability in Eq. (12) to be closed to truth labels with a classification loss. Take multi-class classification as example, we use the cross-entropy loss:

$$\mathcal{L}_{\text{C}} = -\frac{1}{BC} \sum_{i=1}^{B} \sum_{c=1}^{C} \boldsymbol{y}_{ic} \log(p(y = c \mid \hat{\mathbf{z}}_i; \{\boldsymbol{w}^c, \boldsymbol{\mu}^{(c)}\}_{c=1}^{(C)})). \tag{15}$$

With the above loss terms, the final objective of MPHIL can be written as $\mathcal{L} = \mathcal{L}_{\text{C}} + \mathcal{L}_{\text{PS}} + \beta\mathcal{L}_{\text{IPM}}$. The pseudo-code algorithm and complexity analysis of MPHIL is provided in Appendix C.

Table 1: Performance comparison in terms of accuracy. Detailed results with standard deviation are in Table 5 and 6. The best and runner-up results are highlighted in **bold** and underlined. The results of MoleOOD/iMoLD on Motif/CMNIST are not available since they are molecule-specific methods.

| Method | GOOD | | | | | DrugOOD | | | | | |
|---|---|---|---|---|---|---|---|---|---|---|---|
| | Motif | | CMNIST | HIV | | IC50 | | | EC50 | | |
| | basis | size | color | scaffold | size | assay | scaffold | size | assay | scaffold | size |
| ERM | 60.93 | 46.63 | 26.64 | 69.55 | 59.19 | 70.61 | 67.54 | 66.10 | 65.27 | 65.02 | 65.17 |
| IRM | 64.94 | 54.52 | 29.63 | 70.17 | 59.94 | 71.15 | 67.22 | 67.58 | 67.77 | 63.86 | 59.19 |
| VREX | 61.59 | 55.85 | 27.13 | 69.34 | 58.49 | 70.98 | 68.02 | 65.67 | 69.84 | 62.31 | 65.89 |
| Coral | 61.95 | 55.80 | 29.21 | 70.69 | 59.39 | 71.28 | 68.36 | 67.53 | 72.08 | 64.83 | 58.47 |
| MoleOOD | - | - | - | 69.39 | 58.63 | 71.62 | 68.58 | 67.22 | 72.69 | 65.78 | 64.11 |
| CIGA | 67.81 | 51.87 | 25.06 | 69.40 | 61.81 | 71.86 | **69.14** | 66.99 | 69.15 | 67.32 | 65.60 |
| GIL | 65.30 | 54.65 | 31.82 | 68.59 | 60.97 | 70.66 | 67.81 | 66.23 | 70.25 | 63.95 | 64.91 |
| GREA | 59.91 | 47.36 | 22.12 | 71.98 | 60.11 | 70.23 | 67.20 | 66.09 | 74.17 | 65.84 | 61.11 |
| IGM | 74.69 | 52.01 | 33.95 | 71.36 | 62.54 | 68.05 | 63.16 | 63.89 | 76.28 | 67.57 | 60.98 |
| DIR | 64.39 | 43.11 | 22.53 | 68.44 | 57.67 | 69.84 | 66.33 | 62.92 | 65.81 | 63.76 | 61.56 |
| DisC | 65.08 | 42.23 | 23.53 | 58.85 | 49.33 | 61.40 | 62.70 | 64.43 | 63.71 | 60.57 | 57.38 |
| GSAT | 62.27 | 50.03 | 35.02 | 70.07 | 60.73 | 70.59 | 66.94 | 64.53 | 73.82 | 62.65 | 62.65 |
| CAL | 68.01 | 47.23 | 27.15 | 69.12 | 59.34 | 70.09 | 65.90 | 64.42 | 74.54 | 65.19 | 61.21 |
| iMoLD | - | - | - | 72.05 | 62.86 | 71.77 | 67.94 | 66.29 | 77.23 | 66.95 | 67.18 |
| GALA | 72.97 | **60.82** | 40.62 | 71.22 | 65.29 | 70.58 | 66.35 | 66.54 | 77.24 | 66.98 | 63.71 |
| EQuAD | 75.46 | 55.10 | 40.29 | 71.49 | 64.09 | 71.57 | 67.74 | 67.54 | 77.64 | 65.73 | 64.39 |
| MPHIL | **76.23** | 58.43 | **41.29** | **74.69** | **66.84** | **72.96** | 68.62 | **68.06** | **78.08** | **68.94** | **68.11** |

## 4 EXPERIMENTS

In this section, we present our experimental setup (Sec. 4.1) and showcase the results in (Sec. 4.2). For each experiment, we first highlight the research question being addressed, followed by a detailed discussion of the findings.

### 4.1 EXPERIMENTAL SETUP

**Datasets.** We evaluate the performance of MPHIL on two real-world benchmarks, GOOD (Gui et al., 2022) and DrugOOD (Ji et al., 2023), with various distribution shifts to evaluate our method. Specifically, GOOD is a comprehensive graph OOD benchmark, and we selected three datasets: (1) GOOD-HIV (Wu et al., 2018), a molecular graph dataset predicting HIV inhibition; (2) GOOD-CMNIST (Arjovsky et al., 2019), containing graphs transformed from MNIST using superpixel techniques; and (3) GOOD-Motif (Wu et al., 2022b), a synthetic dataset where graph motifs determine the label. DrugOOD is designed for AI-driven drug discovery with three types of distribution shifts: scaffold, size, and assay, and applies these to two measurements (IC50 and EC50). Details of datasets are in Appendix D.1.

**Baselines**. We compare MPHIL against ERM and two kinds of OOD baselines: (1) Traditional OOD generalization approaches, including Coral (Sun & Saenko, 2016), IRM (Arjovsky et al., 2019) and VREx (Krueger et al., 2021); (2) graph-specific OOD generalization methods, including environment-based approaches (MoleOOD (Yang et al., 2022), CIGA (Chen et al., 2022b), GIL (Li et al., 2022c), and GREA (Liu et al., 2022), IGM (Jia et al., 2024)), causal explanation-based approaches (Disc (Fan et al., 2022) and DIR(Wu et al., 2022b)), and advanced architecture-based approaches (CAL (Sui et al., 2022) and GSAT (Miao et al., 2022),iMoLD (Zhuang et al., 2023)),GALA (Yao et al., 2024), EQuAD (Chen et al., 2024). Details of all baselines are in Appendix D.2.

**Implementation Details**. To ensure fairness, we adopt the same experimental setup as iMold across two benchmarks. For molecular datasets with edge features, we use a three-layer GIN with a hidden dimension of 300, while for non-molecular graphs, we employ a four-layer GIN with a hidden dimension of 128. The projector is a two-layer MLP with a hidden dimension set to half that of the GIN encoder. EMA rate $\alpha$ for prototype updating is fixed at 0.99. Adam optimizer is used for model parameter updates. All baselines use the optimal parameters from their original papers. Additional hyperparameter details can be found in Appendix D.3.

### 4.2 PERFORMANCE COMPARISON

In this experiment, we aim to answer **Q1: Whether MPHIL achieves the best performance on OOD generalization benchmarks?** The answer is **YES**, since MPHIL shows the best results on the majority of datasets. Specifically, we have the following observations.

▷ **State-of-the-art results.** According to Table 1, MPHIL achieves state-of-the-art performance on 10 out of 11 datasets, and secures the second place on the remaining dataset. The average improvements against the previous SOTA are 2.17% on GOOD and 1.68% on DrugOOD. Notably, MPHIL achieves competitive performance across various types of datasets with different data shifts, demonstrating its generalization ability on different data. Moreover, our model achieves the best results in both binary and multi-class tasks, highlighting the effectiveness of the multi-prototype classifier in handling different classification tasks.

▷ **Sub-optimal performance of environment-based methods.** Among all baselines, environment-based methods only achieve the best performance on 4 datasets, while other methods perform best on the remaining datasets. Notably, architecture-based OOD generalization methods achieves the best results on the largest number of datasets. These observations suggest that environment-based methods are limited by the challenge of accurately capturing environmental information in graph data, leading to a discrepancy between theoretical expectations and empirical results. In contrast, the remarkable performance of MPHIL also proves that graph OOD generalization can still be achieved without specific environmental information.

### 4.3 ABLATION STUDY

Here, we aim to discover **Q2: Does each module in MPHIL contribute to effective OOD generalization?** The answer is **YES**, as removing any key component leads to performance degradation, as demonstrated by the results in Table 2. We have the following discussions.

▷ **Ablation on $\mathcal{L}_{IPM}$ and $\mathcal{L}_{PS}$.** We remove $\mathcal{L}_{IPM}$ and $\mathcal{L}_{PS}$ in the Eq. (6) respectively to explore their impacts on the performance of OOD generalization. The experimental results demonstrate a clear fact:

Table 2: Performance of MPHIL and its variants.

| Variants | CMNIST-color | HIV-scaffold | IC50-size |
|---|---|---|---|
| MPHIL | **41.29 (3.85)** | **74.69 (1.77)** | **68.06 (0.55)** |
| ERM | 26.64 (2.37) | 69.55 (2.39) | 66.10 (0.31) |
| w/o $\mathcal{L}_{IPM}$ | 37.86 (3.44) | 70.61 (1.52) | 67.09 (0.65) |
| w/o $\mathcal{L}_{PS}$ | 37.53 (2.18) | 71.06 (1.56) | 66.21 (0.37) |
| w/o Project | 21.05 (4.89) | 65.78 (3.57) | 51.96 (2.54) |
| w/o Multi-P | 20.58 (3.78) | 62.11 (1.95) | 57.64 (1.02) |
| w/o Inv. Enc. | 34.86 (2.92) | 66.72 (1.19) | 63.73 (0.89) |
| w/o Update | 38.95 (3.01) | 67.89 (1.84) | 64.14 (1.22) |
| w/o Prune | 40.58 (3.78) | 71.11 (1.95) | 67.64 (1.02) |

merely optimizing for invariance (w/o $\mathcal{L}_{PS}$) or separability (w/o $\mathcal{L}_{IPM}$) weakens the OOD generalization ability of our model, especially for the multi-class classification task. This provides strong evidence that ensuring both invariance and separability is a sufficient and necessary condition for effective OOD generalization in graph learning.

▷ **Ablation on the design of MPHIL.** To verify the effectiveness of each module designed for MPHIL, we conducted ablation studies by removing the hyperspherical projection(w/o Project), multi-prototype mechanism (w/o Multi-P), invariant encoder (w/o Inv.Enc.), and prototype-related weight calculations (w/o Update) and pruning (w/o Prune). The results confirm their necessity. First, removing the hyperspherical projection significantly drops performance, as optimizing Eq. (6) requires hyperspherical space. Without it, results are even worse than ERM. Similarly, setting the prototype count to one blurs decision boundaries and affects the loss function $\mathcal{L}_{PS}$, compromising inter-class separability. Lastly, replacing the invariant encoder $GNN_S$ with $GNN_E$ directly introduces environment-related noise, making it difficult to obtain effective invariant features, thus hindering OOD generalization. Additionally, the removal of prototype-related weight calculations and weight pruning degraded prototypes into the average of all class samples, resulting in the prototypes degrading into the average representation of all samples in the class, failing to maintain classification performance in OOD scenarios.

### 4.4 VISUALIZATION EXPERIMENTS

In this subsection, we aim to investigate **Q3: How can the key designs (i.e., hyperspherical space and multi-prototype mechanism) improve the representation capability of MPHIL from a qualitative perspective?** We conduct the following visualization experiments to answer this question.

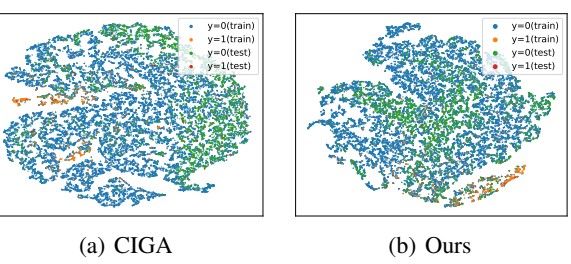

(a) CIGA            (b) Ours

Figure 3: t-SNE visualization on HIV-Scaffold.

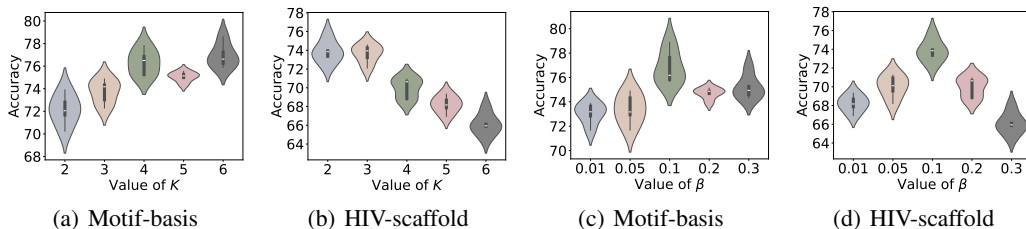

|                |                |                |                |
| :------------: | :------------: | :------------: | :------------: |
| (a) Motif-basis | (b) HIV-scaffold | (c) Motif-basis | (d) HIV-scaffold |

Figure 4: The two figures on the left present a hyperparameter analysis of the number of prototypes $K$, while the two figures on the right illustrate the impact of the coefficient $\beta$ for the loss term $\mathcal{L}_{\text{IPM}}$.

▷ Hyperspherical representation space. To further validate the advantage of hyperspherical latent space over traditional latent spaces, we visualized the invariant representation $\hat{z}_{inv}$ in both the training set (ID) and the test set (OOD). Using t-SNE, we visualize the representation distributions learned by MPHIL and the SOTA method, CIGA, as shown in Fig. 3. It is evident that MPHIL produces more separable invariant representations, while also exhibiting tighter clustering for samples of the same class, indicating successful capture of environment-invariant features. In contrast, although CIGA achieves a certain level of intra-class compactness, its lower separability hinders its overall performance.

▷ Prototypes visualization. We also reveal the characteristics of prototypes by visualizing samples that exhibit the highest similarity to each prototype. Fig. 5 shows that prototypes from different classes capture distinct invariant subgraphs, ensuring a strong correlation with their respective labels. Furthermore, within the same category, different prototypes encapsulate samples with varying environmental subgraphs. This effectively validates the association between multi-prototype learning and environmental adaptability.

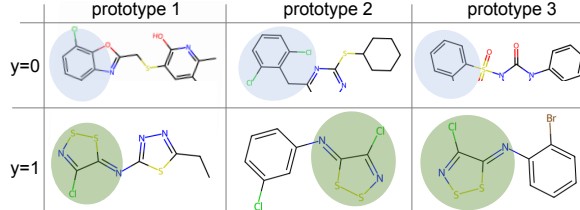

Figure 5: Visualizations of prototypes and invariant subgraphs (highlighted) of IC50-assay dataset.

### 4.5 HYPERPARAMETER ANALYSIS

In this experiment, we will investigate **Q4: How do the key hyperparameters impact the performance of MPHIL?** The following experiments are conducted to answer this question.

▷ Analysis of $K$. To investigate the effect of the number of prototypes on model performance, we vary $k$ from 2 to 6 and present the experimental results in Fig. 4(a) and 4(b). We observe that the best performance is achieved when the number of prototypes is approximately twice the number of classes (e.g., 2 or 3 for GOOD-HIV, and 5 or 6 for GOOD-Motif). Deviating from this optimal range, either too many or too few prototypes negatively impacts the final performance.

▷ Analysis of $\beta$. To discover the sensitivity of MPHIL to coefficient $\beta$ in $\mathcal{L}_{\text{IPM}}$, we search $\beta$ from $\{0.01, 0.05, 0.1, 0.2, 0.3\}$ and present the results in Fig. 4(c) and 4(d). We observe that a small $\beta$ (e.g., 0.01 and 0.05) hampers the model's ability to effectively learn invariant features, while selecting a moderate $\beta$ (i.e., 0.1) leads to the best performance.

## 5 CONCLUSION

In this work, we introduce a novel graph invariant learning framework integrated with hyperspherical space and prototypical learning, ensuring that the learned representations are both environment-invariant and class-separable without relying on environmental information. Building upon this framework, we present a new graph out-of-distribution generalization method named MPHIL. MPHIL achieves inter-class invariance and intra-class separability by optimizing two effective loss functions and leverages class prototypes – defined as the mean feature vectors of each category – to eliminate dependency on individual prototypes. Experimental evaluations on the DrugOOD and GOOD benchmarks demonstrate the effectiveness of MPHIL.

ETHICS STATEMENT

Our research focuses exclusively on scientific questions, with no involvement of human subjects, animals, or environmentally sensitive materials. Therefore, we foresee no ethical risks or conflicts of interest. We are committed to maintaining the highest standards of scientific integrity and ethics to ensure the validity and reliability of our findings.

REPRODUCIBILITY STATEMENT

Our model is clearly formalized in the main text for clarity and comprehensive understanding. Detailed implementation, including related works, proof, metholdogy details, experimental details, is provided in Appendix. The experimental settings and baselines have been rigorously checked for fair comparison. The source code of our method is available in https://anonymous.4open.science/r/MPHIL-23C0/.

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

# A  RELATED WORKS

**Out-of-Distribution (OOD) Generalization.** Due to the sensitivity of deep neural networks to distributional shifts, their performance can vary dramatically, making out-of-distribution (OOD) generalization an important research topic (Arjovsky et al., 2019; Li et al., 2022b; Krueger et al., 2021). OOD generalization is more challenging than domain adaptation because it targets open environments, aiming to generalize from training on known distributions to completely unseen distributions (Farahani et al., 2021; Kundu et al., 2020). The predominant approach for OOD generalization is invariant learning, which encourages the model to learn representations that remain consistent across environments, thus maintaining predictive power on OOD data (Creager et al., 2021). Additionally, methods such as distributionally robust optimization (Rahimian & Mehrotra, 2019), data augmentation (Volpi et al., 2018), and test-time adaptation (Chen et al., 2022a) have also been proposed to tackle the OOD generalization problem.

**Invariant Learning.** Invariant learning explores stable relationships between features and labels across environments, aiming to learn representations that remain effective in OOD scenarios (Creager et al., 2021; Ahuja et al., 2020). Its interpretability is ensured by a causal data generation process (Sun et al., 2020). (Ye et al., 2021) proved the theoretical error lower bound for OOD generalization based on invariant learning. Notably, the definition of environmental information is critical to invariant learning, yet it also restricts its further development, as it often requires the training set to encompass a diverse and comprehensive range of environments (Lin et al., 2022). It has been proved from theoretical and experimental perspectives (Rosenfeld et al., 2020; Nagarajan et al., 2020).

**OOD Generalization on Graphs.** Inspired by invariant learning, many OOD generalization methods in the graph domain have been proposed, adopting this core idea, and several representative works have emerged. (Yang et al., 2022; Li et al., 2022c; Liu et al., 2022; Jia et al., 2024; Fan et al., 2022; Sui et al., 2022; Miao et al., 2022). Their core idea is to design an effective model or learning strategy that can identify meaningful invariant subgraphs from the input while ignoring the influence of environmental noise (Wu et al., 2022b; Chen et al., 2022b). However, the difficulty of modeling environment information in the graph domain has recently garnered attention, with researchers generally agreeing that directly applying invariant learning to graph OOD generalization presents challenges (Chen et al., 2023; Zhuang et al., 2023).

**Hyperspherical Learning.** Hyperspherical learning has gained attention due to its advantages over traditional Euclidean methods in high-dimensional (Davidson et al., 2018; Ke et al., 2022). The core idea lies in using a projector to project representations onto a unit sphere space for prototype-based classification (Mettes et al., 2019). SphereNet first proposed the concept of deep hyperspherical learning based utilized SphereConv as its basic convolution operator, demonstrating that mapping representations onto the hypersphere improves classification accuracy and robustness to variations (Liu et al., 2017). Recently, hyperspherical learning has been extended to applications like contrastive learning and OOD detection, allowing for better disentanglement of features (Ming et al., 2022; Bai et al., 2024). Despite these advancements, challenges remain in effectively combining hyperspherical representations with invariant learning to address OOD generalization in graph-based tasks, where accurately defining environmental information is particularly difficult.

# B  MPHIL OBJECTIVE DEDUCTIONS

## B.1  PROOF OF THE OVERALL OBJECTIVE

In this section, we explain how we derived our goal in Eq. (6) from Eq. (5). Let's recall that Eq. (5) is formulated as:

$$\min_{f_c, g} - I(\boldsymbol{y}; \mathbf{z}_{inv}) + \beta I(\mathbf{z}_{inv}; e), \tag{16}$$

For the first term $-I(\boldsymbol{y}; \mathbf{z}_{inv})$, since we are mapping invariant features to hyperspherical space, we replace $\mathbf{z}_{inv}$ with $\hat{\mathbf{z}}_{inv}$. Then according to the definition of mutual information:

$$-I(\boldsymbol{y}; \hat{\mathbf{z}}_{inv}) = -\mathbb{E}_{\boldsymbol{y}, \hat{\mathbf{z}}_{inv}} \left[ \log \frac{p(\boldsymbol{y}, \hat{\mathbf{z}}_{inv})}{p(\boldsymbol{y}) p(\hat{\mathbf{z}}_{inv})} \right]. \tag{17}$$

We introduce intermediate variables $\boldsymbol{\mu}^y$ to rewrite Eq. (17) as:

$$
\begin{aligned}
-I(\boldsymbol{y};\hat{\mathbf{z}}_{inv}) &= -E_{vy,\hat{\mathbf{z}}_{inv},\boldsymbol{\mu}^y}\left[\log\frac{p(\boldsymbol{y},\hat{\mathbf{z}}_{inv},\boldsymbol{\mu}^y)}{p(\hat{\mathbf{z}}_{inv},\boldsymbol{\mu}^y)p(\boldsymbol{y})}\right] - E_{\boldsymbol{y},\hat{\mathbf{z}}_{inv},\boldsymbol{\mu}^y}\left[\log\frac{p(\boldsymbol{y},\hat{\mathbf{z}}_{inv})p(\hat{\mathbf{z}}_{inv},\boldsymbol{\mu}^y)}{p(\boldsymbol{y},\hat{\mathbf{z}}_{inv},\boldsymbol{\mu}^y)p(\hat{\mathbf{z}}_{inv})}\right] \\
&= -E_{\boldsymbol{y},\hat{\mathbf{z}}_{inv},\boldsymbol{\mu}^y}\left[\log\frac{p(\boldsymbol{y},\hat{\mathbf{z}}_{inv},\boldsymbol{\mu}^y)}{p(\hat{\mathbf{z}}_{inv},\boldsymbol{\mu}^y)p(\boldsymbol{y})}\right] + E_{\boldsymbol{y},\hat{\mathbf{z}}_{inv},\boldsymbol{\mu}^y}\left[\log\frac{p(\boldsymbol{y},\hat{\mathbf{z}}_{inv},\boldsymbol{\mu}^y)p(\hat{\mathbf{z}}_{inv})}{p(\boldsymbol{y},\hat{\mathbf{z}}_{inv})p(\hat{\mathbf{z}}_{inv},\boldsymbol{\mu}^y)}\right] \\
&= -E_{\boldsymbol{y},\hat{\mathbf{z}}_{inv},\boldsymbol{\mu}^y}\left[\log\frac{p(\boldsymbol{y},\hat{\mathbf{z}}_{inv},\boldsymbol{\mu}^y)}{p(\hat{\mathbf{z}}_{inv},\boldsymbol{\mu}^y)p(\boldsymbol{y})}\right] + E_{\boldsymbol{y},\hat{\mathbf{z}}_{inv},\boldsymbol{\mu}^y}\left[\log\frac{p(\boldsymbol{y},\boldsymbol{\mu}^y|\hat{\mathbf{z}}_{inv})}{p(\boldsymbol{y}|\hat{\mathbf{z}}_{inv})p(\boldsymbol{\mu}^y)|\hat{\mathbf{z}}_{inv}}\right].
\end{aligned}
\tag{18}
$$

By the definition of Conditional mutual information, we have the following equation:

$$
\begin{aligned}
-I(\boldsymbol{y};\hat{\mathbf{z}}_{inv}) &= -I(\boldsymbol{y};\hat{\mathbf{z}}_{inv},\boldsymbol{\mu}^y) + I(\boldsymbol{y};\boldsymbol{\mu}^y|\hat{\mathbf{z}}_{inv}), \\
-I(\boldsymbol{y};\boldsymbol{\mu}^y) &= -I(\boldsymbol{y};\hat{\mathbf{z}}_{inv},\boldsymbol{\mu}^y) + I(\boldsymbol{y};\hat{\mathbf{z}}_{inv}|\boldsymbol{\mu}^y).
\end{aligned}
\tag{19}
$$

By merging the same terms, we have:

$$
-I(\boldsymbol{y};\hat{\mathbf{z}}_{inv}) = -I(\boldsymbol{y};\boldsymbol{\mu}^y) + [I(\boldsymbol{y};\boldsymbol{\mu}^y|\hat{\mathbf{z}}_{inv}) - I(\boldsymbol{y};\hat{\mathbf{z}}_{inv}|\boldsymbol{\mu}^y)].
\tag{20}
$$

Since our classification is based on the distance between $\boldsymbol{\mu}^y$ and $\hat{\mathbf{z}}_{inv}$, we add $-I(\boldsymbol{y};\hat{\mathbf{z}}_{inv},\boldsymbol{\mu}^y)$ back into the above equation and obtain a lower bound:

$$
-I(\boldsymbol{y};\hat{\mathbf{z}}_{inv}) \geq -I(\boldsymbol{y};\boldsymbol{\mu}^y) + [I(\boldsymbol{y};\boldsymbol{\mu}^y|\hat{\mathbf{z}}_{inv}) - I(\boldsymbol{y};\hat{\mathbf{z}}_{inv}|\boldsymbol{\mu}^y)] - I(\boldsymbol{y};\hat{\mathbf{z}}_{inv},\boldsymbol{\mu}^y).
\tag{21}
$$

Since the $\boldsymbol{\mu}^y$ are updated by $\hat{\mathbf{z}}_{inv}$ from the same class, we can approximate $I(\boldsymbol{y};\boldsymbol{\mu}^y|\hat{\mathbf{z}}_{inv})$ equal to $I(\boldsymbol{y};\hat{\mathbf{z}}_{inv}|\boldsymbol{\mu}^y)$ and obtain the new lower bound:

$$
-I(\boldsymbol{y};\hat{\mathbf{z}}_{inv}) \geq -I(\boldsymbol{y};\boldsymbol{\mu}^y) - I(\boldsymbol{y};\hat{\mathbf{z}}_{inv},\boldsymbol{\mu}^y).
\tag{22}
$$

For the second term $I(\mathbf{z}_{inv};e)$, we can also rewrite it as:

$$
I(\hat{\mathbf{z}}_{inv};e) = I(\hat{\mathbf{z}}_{inv};e,\boldsymbol{\mu}^y) - I(\hat{\mathbf{z}}_{inv};\boldsymbol{\mu}^y|e).
\tag{23}
$$

Given that the environmental labels $e$ are unknown, we drop the term $I(\hat{\mathbf{z}}_{inv};e,\boldsymbol{\mu}^y)$ as it cannot be directly computed. This leads to the following lower bound:

$$
I(\hat{\mathbf{z}}_{inv};e) \geq -I(\hat{\mathbf{z}}_{inv};\boldsymbol{\mu}^y|e).
\tag{24}
$$

We can obtain a achievable target by Eq. (22) and Eq. (24) as follow:

$$
-I(\boldsymbol{y};\mathbf{z}_{inv}) + \beta I(\mathbf{z}_{inv};e) \geq -I(\boldsymbol{y};\boldsymbol{\mu}^y) - I(\boldsymbol{y};\hat{\mathbf{z}}_{inv},\boldsymbol{\mu}^y) - \beta I(\hat{\mathbf{z}}_{inv};\boldsymbol{\mu}^y|e).
\tag{25}
$$

In fact, $p(\hat{\mathbf{z}}_{inv},\boldsymbol{\mu}^y|e) \geq p(\hat{\mathbf{z}}_{inv};\boldsymbol{\mu}^y)$, Eq. (25) can be achieved by:

$$
-I(\boldsymbol{y};\mathbf{z}_{inv}) + \beta I(\mathbf{z}_{inv};e) \geq -I(\boldsymbol{y};\boldsymbol{\mu}^y) - I(\boldsymbol{y};\hat{\mathbf{z}}_{inv},\boldsymbol{\mu}^y) - \beta I(\hat{\mathbf{z}}_{inv};\boldsymbol{\mu}^y).
\tag{26}
$$

Finally, optimizing Eq. (5) can be equivalent to optimizing its lower bound and we can obtain the objective without environment $e$ as shown in Eq. (6):

$$
\min_{f_c,g} \underbrace{-I(y;\hat{\mathbf{z}}_{inv},\boldsymbol{\mu}^{(y)})}_{\mathcal{L}_{\mathrm{C}}} \underbrace{-I(y;\boldsymbol{\mu}^{(y)})}_{\mathcal{L}_{\mathrm{PS}}} \underbrace{-\beta I(\hat{\mathbf{z}}_{inv};\boldsymbol{\mu}^{(y)})}_{\mathcal{L}_{\mathrm{IPM}}}.
\tag{27}
$$

## B.2 PROOF OF $\mathcal{L}_{\mathrm{C}}$

For the term $I(y;\hat{\mathbf{z}}_{inv},\boldsymbol{\mu}^{(y)})$, it can be written as:

$$
I(y;\hat{\mathbf{z}}_{inv},\boldsymbol{\mu}^{(y)}) = E_{\boldsymbol{y},\hat{\mathbf{z}}_{inv},\boldsymbol{\mu}^y}\left[\log\frac{p(\boldsymbol{y}|\hat{\mathbf{z}}_{inv},\boldsymbol{\mu}^y)}{p(\boldsymbol{y})}\right],
\tag{28}
$$

according to (Seo et al., 2023), we have:

$$I(y; \hat{\mathbf{z}}_{inv}, \boldsymbol{\mu}^{(y)}) \geq E_{\boldsymbol{y}, \hat{\mathbf{z}}_{inv}, \boldsymbol{\mu}^y} \left[ \log \frac{q_\theta(\boldsymbol{y}|\gamma(\hat{\mathbf{z}}_{inv}, \boldsymbol{\mu}^y))}{p(\boldsymbol{y})} \right], \tag{29}$$

where $q_\theta(\boldsymbol{y}|\gamma(\hat{\mathbf{z}}_{inv}, \boldsymbol{\mu}^y))$ is the variational approximation of $p(\boldsymbol{y}|\gamma(\hat{\mathbf{z}}_{inv}, \boldsymbol{\mu}^y))$. $\gamma(,)$ is the function to calculate the similarity between $\hat{\mathbf{z}}_{inv}$ and $\boldsymbol{\mu}^y$. Then we can have:

$$\begin{aligned}
I(y; \hat{\mathbf{z}}_{inv}, \boldsymbol{\mu}^{(y)}) &\geq E_{\boldsymbol{y}, \hat{\mathbf{z}}_{inv}, \boldsymbol{\mu}^y} \left[ \log \frac{q_\theta(\boldsymbol{y}|\gamma(\hat{\mathbf{z}}_{inv}, \boldsymbol{\mu}^y))}{p(\boldsymbol{y})} \right] \\
&\geq E_{\boldsymbol{y}, \hat{\mathbf{z}}_{inv}, \boldsymbol{\mu}^y} [\log q_\theta(\boldsymbol{y}|\gamma(\hat{\mathbf{z}}_{inv}, \boldsymbol{\mu}^y))] - E_{\boldsymbol{y}}[\log p(\boldsymbol{y})] \\
&\geq E_{\boldsymbol{y}, \hat{\mathbf{z}}_{inv}, \boldsymbol{\mu}^y} [\log q_\theta(\boldsymbol{y}|\gamma(\hat{\mathbf{z}}_{inv}, \boldsymbol{\mu}^y))] \\
&:= -\mathcal{L}_{\mathrm{C}}.
\end{aligned} \tag{30}$$

Finally, we prove that $\min I(y; \hat{\mathbf{z}}_{inv}, \boldsymbol{\mu}^{(y)})$ is equivalent to minimizing the classification loss $\mathcal{L}_{\mathrm{C}}$.

## C METHODOLOGY DETAILS

### C.1 OVERALL ALGORITHM OF MPHIL

The training algorithm of MPHIL is shown in Algorithm. 1. After that, we use the well-trained $\mathrm{GNN}_S, \mathrm{GNN}_E$, Proj and all prototypes $\mathbf{M}^{(c)} = \{\boldsymbol{\mu}_k^{(c)}\}_{k=1}^K$ to perform inference on the test set. The pseudo-code for this process is shown in Algorithm. 2.

---

**Algorithm 1** The training algorithm of MPHIL.

---

**Input:** Scoring GNN $\mathrm{GNN}_S$; Encoding GNN $\mathrm{GNN}_E$; Projection Proj; Number of prototypes for each class $K$; The data loader of in-distribution training set $D_{\mathrm{train}}$.
**Output:** Well-trained $\mathrm{GNN}_S$, $\mathrm{GNN}_E$, Proj and all prototypes $\mathbf{M}^{(c)}$.
 1: For each class $c \in \{1, \cdots, C\}$, assign $K$ prototypes for it which can be denoted by $\mathbf{M}^{(c)} = \{\boldsymbol{\mu}_k^{(c)}\}_{k=1}^K$.
 2: Initialize each of them by $\boldsymbol{\mu}_k^{(c)} \sim \mathcal{N}(\mathbf{0}, \mathbf{I})$
 3: **for** epoch in epochs **do**
 4:     **for** each $G_{\mathrm{batch}}$ in $D_{\mathrm{train}}$ **do**
 5:         Obtain $\boldsymbol{Z}_{inv}$ using $\mathrm{GNN}_S$ and $\mathrm{GNN}_E$ via Eq. (7) and (8)
 6:         Obtain $\hat{\boldsymbol{Z}}_{inv}$ using Proj via Eq. (9)
 7:         Compute $\boldsymbol{W}^{(c)}$ using $\boldsymbol{\mu}^{(c)}$ and $\hat{\boldsymbol{Z}}_{inv}$ via Eq. (11).
 8:         **for** each prototype $\boldsymbol{\mu}_k^{(c)}$ **do**
 9:            Update it using $\hat{\boldsymbol{Z}}_{inv}$ and $\boldsymbol{W}^{(c)}$ via Eq. (10).
10:         **end for**
11:         Get $p(y = c \mid \hat{\mathbf{z}}_i; \{\boldsymbol{w}^c, \boldsymbol{\mu}^{(c)}\}_{c=1}^{(C)})$ using $\hat{\boldsymbol{Z}}_{inv}$, $\boldsymbol{W}^{(c)}$ and $\boldsymbol{\mu}^{(c)}$ via Eq. (12)
12:         Compute the final loss $\mathcal{L}$ with $\hat{\boldsymbol{Z}}_{inv}$, $\boldsymbol{\mu}^{(c)}$ and $p(y = c \mid \hat{\mathbf{z}}_i; \{\boldsymbol{w}^c, \boldsymbol{\mu}^{(c)}\}_{c=1}^{(C)})$ via Eq. (13), (14) and (15)
13:         Update parameters of $\mathrm{GNN}_S$, $\mathrm{GNN}_E$ and Proj with the gradient of $\mathcal{L}$.
14:     **end for**
15: **end for**

---

### C.2 WEIGHT PRUNING

Directly assigning weights to all prototypes within a class can lead to excessive similarity between prototypes, especially for difficult samples. This could blur decision boundaries and reduce the model's ability to correctly classify hard-to-distinguish samples.

To address this, we apply a **top-$n$ pruning** strategy, which keeps only the most relevant prototypes for each sample. The max weights are retained, and the rest are pruned as follows:

$$\boldsymbol{W}_{i,k}^{(c)} = \mathbb{1}[\boldsymbol{W}_{i,k}^{(c)} > \beta] * \boldsymbol{W}_{i,k}^{(c)}, \tag{31}$$

---

**Algorithm 2** The inference algorithm of MPHIL.

---

**Input:** Well-trained $\text{GNN}_S$, $\text{GNN}_E$, Proj and all prototypes $\mathbf{M}^{(c)} = \{\boldsymbol{\mu}_k^{(c)}\}_{k=1}^K$. The data loader of Out-of-distribution testing set $D_{\text{test}}$.

**Output:** Classification probability $p(y = c \mid \hat{\mathbf{z}}_i; \{\boldsymbol{w}^c, \boldsymbol{\mu}^{(c)}\}_{c=1}^{(C)})$

1: **for** each $G_{\text{batch}}$ in $D_{\text{test}}$ **do**
2:      Obtain $\boldsymbol{Z}_{inv}$ using $\text{GNN}_S$ and $\text{GNN}_E$ via Eq. (7) and (8)
3:      Obtain $\hat{\boldsymbol{Z}}_{inv}$ using Proj via Eq. (9)
4:      Compute $\boldsymbol{W}^{(c)}$ using $\boldsymbol{\mu}^{(c)}$ and $\hat{\boldsymbol{Z}}_{inv}$ via Eq. (11).
5:      Get $p(y = c \mid \hat{\mathbf{z}}_i; \{\boldsymbol{w}^c, \boldsymbol{\mu}^{(c)}\}_{c=1}^{(C)})$ using $\hat{\boldsymbol{Z}}_{inv}$, $\boldsymbol{W}^{(c)}$ and $\boldsymbol{\mu}^{(c)}$ via Eq. (12)
6: **end for**

---

where $\beta$ is the threshold corresponding to the top-$n$ weight, and $\mathbb{1}[\boldsymbol{W}_{i,k}^{(c)} > \beta]$ is an indicator function that retains only the weights for the top-$n$ prototypes. This pruning mechanism ensures that the prototypes remain distinct and that the decision space for each class is well-defined, allowing for improved classification performance. By applying this attention-based weight calculation and top-$n$ pruning, the model ensures a more accurate and robust matching of samples to prototypes, enhancing classification, especially in OOD scenarios.

### C.3 COMPLEXITY ANALYSIS

The time complexity of MPHIL is $\mathcal{O}(|E|d + |V|d^2)$, where $|V|$ denotes the number of nodes and $|E|$ denotes the number of edges, $d$ is the dimension of the final representation. Specifically, for $\text{GNN}_S$ and $\text{GNN}_E$, their complexity is denoted as $\mathcal{O}(|E|d + |V|d^2)$. The complexity of the projector is $\mathcal{O}(|V|d^2)$, while the complexities of calculating weights and updating prototypes are $\mathcal{O}(|V||K|d)$ where $K$ is the number of prototypes. The complexity of computing the final classification probability also is $\mathcal{O}(|V|Kd)$. Since $K$ is a very small constant, we can ignore $\mathcal{O}(|V|Kd)$, resulting in a final complexity of $\mathcal{O}(|E|d + |V|d^2)$. Theoretically, the time complexity of MPHIL is on par with the existing methods.

## D EXPERIMENTAL DETAILS

### D.1 DATASETS

**Overview of the Dataset.** In this work, we use 11 publicly benchmark datasets, 5 of them are from GOOD (Gui et al., 2022) benchmark. They are the combination of 3 datasets (GOOD-HIV, GOOD-Motif and GOOD-CMNIST) with different distribution shift (scaffold, size, basis, color). The rest 6 datasets are from DrugOOD (Ji et al., 2023) benchmark, including IC50-Assay, IC50-Scaffold, IC50-Size, EC50-Assay, EC50-Scaffold, and EC50-Size. The prefix denotes the measurement and the suffix denotes the distribution-splitting strategies. We use the default dataset split proposed in each benchmark. Statistics of each dataset are in Table 3.

**Distribution split.** In this work, we investigate various types of distribution-splitting strategies for different datasets.

- **Scaffold.** Molecular scaffold is the core structure of a molecule that supports its overall composition, but it only exhibits specific properties when combined with particular functional groups. Distribution shift occurs when there are significant changes in the scaffold structure.

- **Size.** The size of a graph refers to the total number of nodes, and it is also implicitly related to the graph's structural properties. Distribution shift occurs when there is a significant change in graph size.

- **Assay.** The assay is an experimental technique used to examine or determine molecular characteristics. Due to differences in assay conditions and targets, activity values measured by different assays can vary significantly. Assay variation results in a distribution shift.

Table 3: Dateset statistics.

| Dataset | | | Task | Metric | Train | Val | Test |
|---|---|---|---|---|---|---|---|
| GOOD | HIV | scaffold | Binary Classification | ROC-AUC | 24682 | 4133 | 4108 |
| | | size | Binary Classification | ROC-AUC | 26169 | 4112 | 3961 |
| | Motif | basis | Multi-label Classification | ACC | 18000 | 3000 | 3000 |
| | | size | Multi-label Classification | ACC | 18000 | 3000 | 3000 |
| | CMNIST | color | Multi-label Classification | ACC | 42000 | 7000 | 7000 |
| DrugOOD | IC50 | assay | Binary Classification | ROC-AUC | 34953 | 19475 | 19463 |
| | | scaffold | Binary Classification | ROC-AUC | 22025 | 19478 | 19480 |
| | | size | Binary Classification | ROC-AUC | 37497 | 17987 | 16761 |
| | EC50 | assay | Binary Classification | ROC-AUC | 4978 | 2761 | 2725 |
| | | scaffold | Binary Classification | ROC-AUC | 2743 | 2723 | 2762 |
| | | size | Binary Classification | ROC-AUC | 5189 | 2495 | 2505 |

- **Basis.** The generation of a motif involves combining a base graph (wheel, tree, ladder, star, and path) with a motif (house, cycle, and crane), but only the motif is directly associated with the label. Distribution shift occurs when the base graph changes.

- **Color.** CMNIST is a graph dataset constructed from handwritten digit images. Following previous research, we declare a distribution shift when the color of the handwritten digits changes.

## D.2 BASELINES

In our experiments, the methods we compared can be divided into two categories, one is ERM and traditional OOD generalization methods:

- **ERM** is a standard learning approach that minimizes the average training error, assuming the training and test data come from the same distribution.

- **IRM** (Arjovsky et al., 2019) aims to learn representations that remain invariant across different environments, by minimizing the maximum error over all environments.

- **VREx** (Krueger et al., 2021) propose a penalty on the variance of training risks which can providing more robustness to changes in the input distribution.

- **Coral** (Sun & Saenko, 2016) utilize a nonlinear transformation to align the second-order statistical features of the source and target domain distributions

Another class of methods is specifically designed for Graph OOD generalization:

- **MoleOOD** (Yang et al., 2022) learn the environment invariant molecular substructure by a environment inference model and a molecular decomposing model.

- **CIGA** (Chen et al., 2022b) proposes an optimization objective based on mutual information to ensure the learning of invariant subgraphs that are not affected by the environment.

- **GIL** (Li et al., 2022c) performs environment identification and invariant risk loss optimization by separating the invariant subgraph and the environment subgraph.

- **GERA** (Liu et al., 2022) performs data augmentation by replacing the input graph with the environment subgraph to improve the generalization ability of the model

- **IGM** (Jia et al., 2024) performs data augmentation by simultaneously performing a hybrid strategy of invariant subgraphs and environment subgraphs.

- **DIR** (Wu et al., 2022b) identifies causal associations between input graphs and labels by performing counterfactual interventions.

Table 4: Hyper-parameter configuration.

| | | | proj_dim | att_dim | $K$ | $lr$ | $\beta$ |
|---|---|---|---|---|---|---|---|
| DrugOOD | IC50 | Assay | 300 | 128 | 3 | 0.001 | 0.1 |
| | | Scaffold | 300 | 128 | 3 | 0.001 | 0.1 |
| | | Size | 300 | 128 | 3 | 0.001 | 0.1 |
| | EC50 | Assay | 300 | 128 | 3 | 0.001 | 0.1 |
| | | Scaffold | 300 | 128 | 3 | 0.001 | 0.1 |
| | | Size | 300 | 128 | 3 | 0.001 | 0.1 |
| GOOD | HIV | Scaffold | 300 | 128 | 3 | 0.01 | 0.1 |
| | | Size | 300 | 128 | 3 | 0.01 | 0.1 |
| | Motif | Basis | 256 | 128 | 6 | 0.01 | 0.2 |
| | | Size | 256 | 128 | 6 | 0.01 | 0.2 |
| | CMNIST | Color | 256 | 128 | 5 | 0.01 | 0.2 |

- **DisC** (Fan et al., 2022) learns causal and bias representations through a causal and disentangling based learning strategy separately.
- **GSAT** (Miao et al., 2022) learns the interpretable label-relevant subgraph through an attention mechanism that is injected with stochasticity.
- **CAL** Sui et al. (2022) proposes a causal attention learning strategy to ensure that GNNs learn effective representations instead of optimizing loss through shortcuts.
- **iMoLD** (Zhuang et al., 2023) designs two GNNs to directly extract causal features from the encoded graph representation.
- **GALA** (Chen et al., 2024) designs designs a new loss function to ensure graph OOD generalization without environmental information as much as possible.
- **EQuAD** (Yao et al., 2024) learns how to effectively remove spurious features by optimizing the self-supervised informax function.

### D.3 IMPLEMENTATION DETAILS

**Baselines.** For all traditional OOD methods, we conduct experiments on different datasets using the code provided by GOOD (Gui et al., 2022) and DrugOOD (Ji et al., 2023) benchmark. For graph OOD generalization methods with public code, we perform experiments in the same environments as our method and employ grid search to select hyper-parameters, ensuring fairness in the results.

**Our method.** We implement our proposed MPHIL under the Pytorch (Paszke et al., 2019) and PyG (Fey & Lenssen, 2019). For all datasets containing molecular graphs (all datasets from DrugOOD and GOODHIV), we fix the learning rate to $0.001$ and select the hyper-parameters by ranging the $\text{proj\_dim}$ from $\{100, 200, 300\}$, $\text{att\_dim}$ from $\{64, 128, 256\}$, $K$ from $\{2, 3, 4, 5\}$ and $\beta$ from $\{0.01, 0.1, 0.2\}$. For the other datasets, we fix the learning rate to $0.01$ and select the hyper-parameters by ranging the $\text{proj\_dim}$ from $\{64, 128, 256\}$, $\text{att\_dim}$ from $\{64, 128, 256\}$, $K$ from $\{3, 4, 5, 6\}$ and $\beta$ from $\{0.01, 0.1, 0.2\}$. For the top-$n$ pruning, we force $n$ to be half of $K$. We conduct a grid search to select hyper-parameters and refer to Table 4 for the detailed configuration. For all experiments, we fix the number of epochs to 200 and run the experiment five times with different seeds, select the model to run on the test set based on its performance on validation, and report the mean and standard deviation.

## D.4 SUPPLEMENTAL RESULTS

We report the complete experimental results with means and standard deviations in Tables 5 and 6 .

Table 5: Performance comparison in terms of average accuracy (standard deviation) on GOOD benchmark.

| Method | GOOD-Motif | | GOOD-CMNIST | GOOD-HIV | |
| --- | --- | --- | --- | --- | --- |
| | basis | size | color | scaffold | size |
| ERM | 60.93 (2.11) | 46.63 (7.12) | 26.64 (2.37) | 69.55 (2.39) | 59.19 (2.29) |
| IRM | 64.94 (4.85) | 54.52 (3.27) | 29.63 (2.06) | 70.17 (2.78) | 59.94 (1.59) |
| VREX | 61.59 (6.58) | 55.85 (9.42) | 27.13 (2.90) | 69.34 (3.54) | 58.49 (2.28) |
| Coral | 61.95 (4.36) | 55.80 (4.05) | 29.21 (6.87) | 70.69 (2.25) | 59.39 (2.90) |
| MoleOOD | - | - | - | 69.39 (3.43) | 58.63 (1.78) |
| CIGA | 67.81 (2.42) | 51.87 (5.15) | 25.06 (3.07) | 69.40 (1.97) | 61.81 (1.68) |
| GIL | 65.30 (3.02) | 54.65 (2.09) | 31.82 (4.24) | 68.59 (2.11) | 60.97 (2.88) |
| GREA | 59.91 (2.74) | 47.36 (3.82) | 22.12 (5.07) | 71.98 (2.87) | 60.11 (1.07) |
| IGM | 74.69 (8.51) | 52.01 (5.87) | 33.95 (4.16) | 71.36 (2.87) | 62.54 (2.88) |
| DIR | 64.39 (2.02) | 43.11 (2.78) | 22.53 (2.56) | 68.44 (2.51) | 57.67 (3.75) |
| DisC | 65.08 (5.06) | 42.23 (4.20) | 23.53 (0.67) | 58.85 (7.26) | 49.33 (3.84) |
| GSAT | 62.27 (8.79) | 50.03 (5.71) | 35.02 (2.78) | 70.07 (1.76) | 60.73 (2.39) |
| CAL | 68.01 (3.27) | 47.23 (3.01) | 27.15 (5.66) | 69.12 (1.10) | 59.34 (2.14) |
| GALA | 66.91 (2.77) | 45.39 (5.84) | 38.95 (2.97) | 69.12 (1.10) | 59.34 (2.14) |
| iMoLD | - | - | - | 72.05 (2.16) | 62.86 (2.34) |
| GALA | 72.97 (4.28) | **60.82 (0.51)** | 40.62 (2.11) | 71.22 (1.93) | 65.29 (0.72) |
| EQuAD | 75.46 (4.35) | 55.10 (2.91) | 40.29 (3.95) | 71.49 (0.67) | 64.09 (1.08) |
| MPHIL | **76.23 (4.89)** | 58.43 (3.15) | **41.29 (3.85)** | **73.94 (1.77)** | **66.84 (1.09)** |

Table 6: Performance comparison in terms of average accuracy (standard deviation) on DrugOOD benchmark.

| Method | DrugOOD-IC50 | | | DrugOOD-EC50 | | |
| --- | --- | --- | --- | --- | --- | --- |
| | assay | scaffold | size | assay | scaffold | size |
| ERM | 70.61 (0.75) | 67.54 (0.42) | 66.10 (0.31) | 65.27 (2.39) | 65.02 (1.10) | 65.17 (0.32) |
| IRM | 71.15 (0.57) | 67.22 (0.62) | 67.58 (0.58) | 67.77 (2.71) | 63.86 (1.36) | 59.19 (0.83) |
| VREx | 70.98 (0.77) | 68.02 (0.43) | 65.67 (0.19) | 69.84 (1.88) | 62.31 (0.96) | 65.89 (0.83) |
| Coral | 71.28 (0.91) | 68.36 (0.61) | 67.53 (0.32) | 72.08 (2.80) | 64.83 (1.64) | 58.47 (0.43) |
| MoleOOD | 71.62 (0.50) | 68.58 (1.14) | 67.22 (0.96) | 72.69 (4.16) | 65.78 (1.47) | 64.11 (1.04) |
| CIGA | 71.86 (1.37) | **69.14 (0.70)** | 66.99 (1.40) | 69.15 (5.79) | 67.32 (1.35) | 65.60 (0.82) |
| GIL | 70.66 (1.75) | 67.81 (1.03) | 66.23 (1.98) | 70.25 (5.79) | 63.95 (1.17) | 64.91 (0.76) |
| GREA | 70.23 (1.17) | 67.20 (0.77) | 66.09 (0.56) | 74.17 (1.47) | 65.84 (1.35) | 61.11 (0.46) |
| IGM | 68.05 (1.84) | 63.16 (3.29) | 63.89 (2.97) | 76.28 (4.43) | 67.57 (0.62) | 60.98 (1.05) |
| DIR | 69.84 (1.41) | 66.33 (0.65) | 62.92 (1.89) | 65.81 (2.93) | 63.76 (3.22) | 61.56 (4.23) |
| DisC | 61.40 (2.56) | 62.70 (2.11) | 64.43 (0.60) | 63.71 (5.56) | 60.57 (2.27) | 57.38 (2.48) |
| GSAT | 70.59 (0.43) | 66.94 (1.43) | 64.53 (0.51) | 73.82 (2.62) | 62.65 (1.79) | 62.65 (1.79) |
| CAL | 70.09 (1.03) | 65.90 (1.04) | 64.42 (0.50) | 74.54 (1.48) | 65.19 (0.87) | 61.21 (1.76) |
| iMoLD | 71.77 (0.54) | 67.94 (0.59) | 66.29 (0.74) | 77.23 (1.72) | 66.95 (1.26) | 67.18 (0.86) |
| GALA | 70.58 (2.63) | 66.35 (0.86) | 66.54 (0.93) | 77.24 (2.17) | 66.98 (0.84) | 63.71 (1.17) |
| EQuAD | 71.57 (0.95) | 67.74 (0.57) | 67.54 (0.27) | 77.64 (0.63) | 65.73 (0.17) | 64.39 (0.67) |
| MPHIL | **72.96 (1.21)** | 68.62 (0.78) | **68.06 (0.55)** | **78.08 (0.54)** | **68.34 (0.61)** | **68.11 (0.58)** |

