# OpenReview forum: "MPHIL: Multi-Prototype Hyperspherical Invariant Learning for Graph Out-of-Distribution Generalization"
_ICLR.cc/2025/Conference — Submitted to ICLR 2025_

### Official Review · Reviewer_s53C · 2024-10-17

**Soundness:** 3
**Presentation:** 3
**Contribution:** 3
**Rating:** 6
**Confidence:** 3

**Summary:**

This paper introduces MPHIL, a novel method for graph out-of-distribution (OOD) generalization. Key features include: 1）Invariant learning in hyperspherical space. 2) Multi-prototype classification approach. MPHIL eliminates the need for explicit environment modeling and addresses the semantic cliff issue. It outperforms existing methods on 11 OOD benchmark datasets across various domains. The approach offers a more robust solution for real-world graph learning applications where data distributions often shift.

**Strengths:**

1. Originality: The idea is relatively novel, using Hyperspherical Learning to deal with OOD generalization on graphs, and achieving promising experimental results.

2. Clarity & Significance: The logic of the manuscript is clear. In particular, the example of molecular graphs given in the introduction is very helpful for contextualization, and does a good job of describing the problem the authors were trying to solve and the significance of the study.

3. The manuscript is substantial and essentially encompasses all of the research that should be discussed.

**Weaknesses:**

1. There are some terms and formulas that are not presented accordingly, such as I(z;y) in Eq. 2, which I understand to mean mutual information. However, to increase readability I would suggest that these symbols should be explained the first time they appear. Apart from this, there is no explanation of the significance of some of the designed variables, such as separation score S. Why add it additionally to H? What is the benefit of doing so?

2. Figure 3 does not significantly show the advantage of your proposed method over baseline, can you consider a different visualization example that more intuitively reflects the difference between the two?

3.  You included a link to the code in your manuscript, but I found that many of the modules are called, but the relevant code doesn't exist and doesn't work.

**Questions:**

1. Is graph invariant learning only applicable to graph classification task? Can it be processed if it is a node classification task?

2. In Section 3.3, it is mentioned that K prototypes are assigned to each category, so what is the basis for the final categorization? Further, since invariant representation can correspond to a set of prototypes, does it happen that these prototypes do not belong to the same class at the test phase? (after softmax operation, although one of the prototypes has the highest probability, but the other classes contain multiple prototypes with relatively high probability, can it be considered that the confidence of the classification result is lower?) I'm a bit confused here, can you explain the rationale for designing prototypes? Is it relying exclusively on the several losses that have been subsequently proposed to be corrected to avoid the situation I have described?  (I completely agree with your statement at the beginning of section 3.3 about the risk of overfitting caused by a single prototype in the previous approach)

---

> ### Author Response · Authors · 2024-11-19
> **Response to Reviewer  s53C (1/3)**
>
> **W1: There are some terms and formulas that are not presented accordingly, such as I(z;y) in Eq. 2, which I understand to mean mutual information. However, to increase readability I would suggest that these symbols should be explained the first time they appear.**
>
> **A1:** Thanks for your valuable comments, and we sincerely apologize for overlooking the explanation of some essential terms and formulas. In the revised manuscript, we have included the necessary clarifications (page 3 line 160). As you correctly pointed out, $I(z;y)$ represents the mutual information between two random variables, quantifying the amount of shared information between them.
>
> **W2: Apart from this, there is no explanation of the significance of some of the designed variables, such as separation score S. Why add it additionally to H? What is the benefit of doing so?**
>
> **A2:** Thanks for your valuable question! We designed the separation score to be more efficient in obtaining the invariant representation. Most existing methods aim to obtain invariant representations from the input graph by learning a mask matrix to obtain invariant subgraph and encode it, which is time-consuming and inefficient. To address this limitation, we encode the entire graph directly into a vector representation and then utilize the separability score $S$ to recover the invariant representation. This design not only improves computational efficiency but also ensures the accuracy of the extracted invariant representations, as evidenced by the ablation study in Table 2 (w/o Inv. Enc.).
>
> **W3: Figure 3 does not significantly show the advantage of your proposed method over baseline, can you consider a different visualization example that more intuitively reflects the difference between the two?**
>
> **A3:** Thanks for your valuable question! In the GOODHIV dataset, which is a binary classification task, the blue and green points represent class 0 samples from the training and test sets after dimensionality reduction with t-SNE, while the orange and red points represent class 1 samples. We observe that, with MPHIL, the representations of the same class (blue and green, or orange and red) are more tightly clustered, and the separation between different classes (blue vs. orange, and green vs. red) is larger, whether for  In-distribution(training set) or Out of-distribution (test set) data.
>
> We acknowledge that Fig. 3 may be challenging to interpret due to the imbalance in the number of point, as training sets are generally much larger than test sets, especially in out-of-distribution generalization tasks. To address this and further quantify MPHIL's advantages, we calculated the 1-order Wasserstein distance [1] between samples of the same category and those of different categories. The results clearly demonstrate the effectiveness of our method in achieving the principles of 'intra-class compactness and inter-class separation.
>
> | methods| y = 0 intra-class distance $\downarrow$| y = 1 intra-class distance$\downarrow$|inter-class distance $\uparrow$|
> |   :----:|   :---: |     :---: |   :---: |
> |**MPHIL**| **0.17** | **0.52** |**1.09** |
> |SOTA| 0.22 | 0.71 |0.63 |
>
> [1] Villani C. Optimal transport: old and new[M]. Berlin: springer, 2009.
>
> **W4:You included a link to the code in your manuscript, but I found that many of the modules are called, but the relevant code doesn't exist and doesn't work.**
>
> **A4:** Thanks for your valuable question! We apologize for not clearly explaining how to use the code provided in the anonymous link. Some modules that are referenced but not included in the files may need to be installed via pip. To address this, we have added a ```requirements.txt ``` file to the anonymous GitHub repository, specifying all the necessary dependencies. In addition, installation descriptions of the two benchmarks covered in this article have been added to the latest ```Readme``` file.

---

> > ### Author Response · Authors · 2024-11-19
> > **Response to Reviewer s53C (2/3)**
> >
> > **Q1:Is graph invariant learning only applicable to graph classification task? Can it be processed if it is a node classification task?**
> >
> > **A1:** Thanks for your valuable question! Graph invariant learning (GIL) can indeed be extended to node classification tasks, as shown in [1, 2, 3]. However, the workflow differs from its application to graph classification. For graph classification, GIL primarily focuses on extracting a subgraph related to the graph-level label as the invariant representation. In contrast, node classification requires learning a specific invariant representation for each node, emphasizing the construction of an invariant ego-subgraph tailored to each node. While in this paper, like most of the related work, focuses on address the OOD generalization problem at the graph level, extending this approach to node-level tasks is indeed valuable and is the direction we plan to explore further in the future.
> >
> > [1] Li H, Zhang Z, Wang X, et al. Invariant node representation learning under distribution shifts with multiple latent environments[J]. ACM Transactions on Information Systems, 2023, 42(1): 1-30.
> >
> > [2] Wang Q, Wang Y, Wang Y, et al. Dissecting the Failure of Invariant Learning on Graphs[J]. arXiv preprint arXiv:2411.02847, 2024.
> >
> > [3] Liu Y, Ao X, Feng F, et al. FLOOD: A flexible invariant learning framework for out-of-distribution generalization on graphs[C]//Proceedings of the 29th ACM SIGKDD Conference on Knowledge Discovery and Data Mining. 2023: 1548-1558.
> >
> > **Q2: In Section 3.3, it is mentioned that K prototypes are assigned to each category, so what is the basis for the final categorization?**
> >
> > **A2:** Thanks for your valuable question!  As described in Equation 12, each class is associated with $K$ prototypes. To determine the classification probability of a sample for class $c$, we compute the weighted similarities between the sample and the $K$ prototypes of class $c$. The **maximum similarity value** is then used as the final probability for determining whether the sample belongs to class $c$.
> >
> > **Q3: Further, since invariant representation can correspond to a set of prototypes, does it happen that these prototypes do not belong to the same class at the test phase?
> > (after softmax operation, although one of the prototypes has the highest probability, but the other classes contain multiple prototypes with relatively high probability, can it be considered that the confidence of the classification result is lower?)**
> >
> > **A3:** Thanks for your valuable question! The design of multiple prototypes is aimed at achieving **more robust classification**. It is **unrealistic** to expect a single prototype to represent all samples within a class accurately, especially when outliers or slightly deviated samples are present. Our prototype update formula (Eq 10) ensures that each prototype can capture samples located in different regions of the feature space for the same class.
> >
> > Therefore, it is indeed possible for a sample to exhibit **relative low similarity** with certain prototypes of its correct class, as those prototypes are not specifically designed to capture that sample. However, as mentioned in A2, we employ a max operation to select the prototype with the **highest similarity** for the sample, ensuring that it is **accurately captured** by one of the prototypes within its correct class. This facilitates **faster convergence** of the sample towards its correct class space. Importantly, this does not imply lower classification confidence, as the final classification probability is always determined by the highest similarity with the correct class’s prototypes. By optimizing the proposed two loss functions, we ensure that the prototypes of the correct class consistently exhibit higher similarity to the sample compared to prototypes of other classes, thereby guaranteeing high classification confidence.

---

> ### Author Response · Authors · 2024-11-19
> **Response to Reviewer s53C (3/3)**
>
> **Q4: I'm a bit confused here, can you explain the rationale for designing prototypes? Is it relying exclusively on the several losses that have been subsequently proposed to be corrected to avoid the situation I have described?**
>
> **A4:** Thanks for your valuable question! As we explained in A3, the introduction of multiple prototypes is aimed at achieving more robust classification. Beyond that, in our formulation, the misalignment of a sample with an incorrect prototype can be interpreted as **a signal of environmental interference**, while successful alignment with the correct prototype indicates the capture of **stable and invariant features**. By optimizing the two proposed loss functions, we ensure:
> - Each prototype within a class captures a sufficient number of samples, guaranteeing accurate classification into the correct class.
> - Prototypes of different classes are as far apart as possible, maximizing inter-class separability and enabling highly confident classifications.
>
> In summary, by optimizing the two proposed loss functions, we can effectively avoid the classification confidence issue you mentioned, as it ensures that all prototypes of the correct class consistently exhibit **higher classification confidence** compared to prototypes of other classes.

---

> > ### Comment · Reviewer_s53C · 2024-11-19
> >
> > Thank you for your attentive reply which addressed most of my concerns, I will boost my score to 6. Good luck!

---

> > > ### Author Response · Authors · 2024-11-19
> > > **Thanks Reviewer s53C's comments**
> > >
> > > Thank you sincerely for your willingness to raise your score and acknowledge our work. We will make improvements to the manuscript based on your valuable feedback.

---

### Official Review · Reviewer_pafp · 2024-11-01

**Soundness:** 2
**Presentation:** 3
**Contribution:** 2
**Rating:** 5
**Confidence:** 3

**Summary:**

This paper studies the problem of graph Out-of-distribution (OOD) generalization and then proposes two objective functions: the invariant prototype matching loss to ensure samples are matched to the correct class prototypes, and the prototype separation loss to increase the distinction between prototypes of different classes in the hyperspherical space. Extensive experiments on 11 OOD generalization benchmark datasets demonstrate the effectiveness of the proposed method.

**Strengths:**

- The paper studies an important research question.
- The experimental results are extensive.

**Weaknesses:**

- This paper seems to closely follow the recent research "HYPO: Hyperspherical Out-Of-Distribution Generalization", e.g., "intra-class variation and inter-class separation principles". The novelty of the paper seems to be limited.
-  The visualization results do not seem clear (fig 3). Why the proposed method is better than the compared method
- Some SOTA methods seem to be missing.

**Questions:**

See above.

---

> ### Author Response · Authors · 2024-11-19
> **Response to Reviewer pafp (1/2)**
>
> **Q1: This paper seems to closely follow the recent research "HYPO: Hyperspherical Out-Of-Distribution Generalization", e.g., "intra-class variation and inter-class separation principles". The novelty of the paper seems to be limited.**
>
> **A1:** Thanks for your valuable comments! We would like to further clarify the novelty of our work and highlight the differences with the HYPO method.
>
> **1.Novelty of Motivation.** Our work is not a simple application of hyperspherical space but rather an attempt to address two specific challenges in graph OOD generalization: **the difficulty in capturing environmental information** and **the semantic cliff across different classes**. HYPO, on the other hand, **is applied to image data**, where accurate environmental information is accessible. Therefore, the challenges and motivations behind our work differ significantly from those of HYPO.
>
> **2.Novelty of Methodology.** Since reliance on environmental information is not feasible, specific adjustments are made in the graph OOD generalization approach to implement the 'intra-class variation and inter-class separation principles.' Specifically, the MPHIL framework proposed in this work consists of two novel components:
> - **Hyperspherical Invariant Representation Learning (Section 3.2):** This component improves the separability and informativeness of the learned invariant representations, making it more effective at capturing invariant features for graph OOD tasks compared to original representation learning methods.
> - **Multi-Prototype-based Classification (Section 3.3):** This component eliminates the need for explicit environmental modeling and enables more flexible decision boundaries, allowing for better prediction.
>
> These components work together and optimize two novel objective functions: the **invariant prototype matching loss** and the **prototype separation loss** to ensure correct prototype matching and maximize class separability in the hyperspherical space.
> Through these innovative components, our method provides a new perspective and solution for graph OOD generalization, setting it apart from existing methods in computer vision like HYPO in terms of goals and implementation strategies.
>
> **Q2:The visualization results do not seem clear (fig 3). Why the proposed method is better than the compared method.**
>
> **A2:** Thanks for your valuable comments! To visualize the learned representations, we use t-SNE for dimensionality reduction. In the GOODHIV dataset, which is a binary classification task, the blue and green points represent class 0 samples from the training and test sets, while the orange and red points represent class 1 samples. We observe that, with MPHIL, the representations of the same class (blue and green, or orange and red) are more tightly clustered, and the separation between different classes (blue vs. orange, and green vs. red) is larger, both in the training and test sets.
>
> We acknowledge that Fig. 3 seems to be challenging to understand due to the imbalance in the number of points, as training sets are typically much larger than test sets, especially in out-of-distribution generalization tasks. To further quantify the advantage of MPHIL, we also calculate the 1-order Wasserstein distance [1] of the same category and different categories. The result demonstrates the effectiveness of our approach in achieving 'intra-class variation and inter-class separation  principles'.
>
> | methods| y = 0 intra-class distance $\downarrow$| y = 1 intra-class distance$\downarrow$|inter-class distance $\uparrow$|
> |   :----:|   :---: |     :---: |   :---: |
> |**MPHIL**| **0.17** | **0.52** |**1.09** |
> |SOTA| 0.22 | 0.71 |0.63 |
>
>  [1] Villani C. Optimal transport: old and new[M]. Berlin: springer, 2009.

---

> > ### Author Response · Authors · 2024-11-19
> > **Response to Reviewer pafp (2/2)**
> >
> > **Q3: Some SOTA methods seem to be missing.**
> >
> > **A3:** Thanks for your valuable comments! We have added three recent graph out-of-distribution generalization methods: EQuAD [1], LECI [2], and GALA [3] to our baseline comparisons. We reproduced their results using the official code provided by the authors and followed our experimental setup. We present the experimental results in the following table.
> >
> > The results demonstrate that MPHIL consistently achieves **the best average performance**. While LECI shows superior performance on certain datasets, it depends on accurate environment segmentation, which is challenging in real-world scenarios. In contrast, MPHIL does not require additional environmental information and introduces the **Hyperspherical Invariant Representation Learning** and **multi-prototype classification mechanism** to ensure that the learned features are **invariant across environments** while maintaining **maximal class separability**.
> >
> > | Methods| Motif-basis| Motif-size| CMNIST-color| HIV-scaffold|  HIV-size|
> > |   :----:|   :---: |     :---: |   :---: | :---: |:---: |
> > |**MPHIL**| **76.23±4.89** | 58.43±3.15 |41.29±3.85 |**73.94±1.77** |**66.84±1.09** |
> > |EQuAD| 75.46±4.35 | 55.10±2.91 | 40.29±3.95 | 71.49±0.67 | 64.09±1.08 |
> > |LECI| 73.65±2.21 | **62.18±1.84** | **42.88±2.61** | 72.01±0.91 | 63.12±1.62 |
> > |GALA| 72.97±4.28 | 60.82±0.51| 40.62±2.11| 71.22±1.93| 65.29±0.72|
> >
> > | Methods| Ic50-assay| Ic50-scaffold| Ic50-size|Ec50-assay| Ec50-scaffold| Ec50-size
> > |   :----:|   :---: |     :---: |   :---: | :---: |:---: |:---: |
> > |**MPHIL**| **72.96±1.21** | **68.62±0.78** |**68.06±0.55** |**78.08±0.54** |**68.34±0.61** | **68.11±0.58** |
> > |EQuAD| 71.57±0.95 | 67.74±0.57 | 67.54±0.27 | 77.64±0.63 | 65.73±0.17 | 64.39±0.67 |
> > |LECI| 71.57±0.92 | 65.01±0.78 | 64.79±0.86 | 73.51±0.62 | 62.89±0.82 | 63.47±0.95 |
> > |GALA| 70.58±2.63 | 66.35±0.86| 66.54±0.93| 77.24±2.17| 66.98±0.84|63.71±1.17 |
> >
> > [1] Yao T, Chen Y, Chen Z, et al. Empowering graph invariance learning with deep spurious infomax[J]. arXiv preprint arXiv:2407.11083, 2024.
> >
> > [2] Gui S, Liu M, Li X, et al. Joint learning of label and environment causal independence for graph out-of-distribution generalization[J]. Advances in Neural Information Processing Systems, 2024, 36.
> >
> > [3] Chen Y, Bian Y, Zhou K, et al. Does invariant graph learning via environment augmentation learn invariance?[J]. Advances in Neural Information Processing Systems, 2024, 36.

---

> ### Author Response · Authors · 2024-11-22
> **Invitation to Discussion**
>
> Dear Reviewer pafp:
>
> We sincerely appreciate your valuable contributions during the review process and your recognition of and suggestions for our work. To save you time, we provide a **summarized version** of our response to facilitate a quicker understanding:
>
> - **Emphasis on Novelty**: To emphasize the distinctions between our approach and vision-domain methods like HYPO, we address both the motivation and design. Unlike visual OOD generalization, graph OOD generalization faces two unique challenges: **the difficulty in capturing environmental information** and the **semantic cliff across different classes**.  To tackle these issues, we propose two innovative modules: **hyperspherical invariant learning** and **multi-prototype-based classification**, combined with **two novel loss functions**,  enabling **maximally class-separable, environment-free graph OOD generalization**
>
> - **Supplemental Baseline Models**: We have added three state-of-the-art graph OOD generalization methods for comparison, as suggested. The experimental results **continue to highlight the superior performance of MPHIL**.
>
> - **The Interpretation of Figure 3**: We have refined our interpretation of Figure 3 and included **additional quantitative analyses** to better demonstrate the advantages of MPHIL.
>
> We have carefully **response all your questions** and truly appreciate the opportunity to clarify and improve our work based on your feedback.  With some time remaining before the rebuttal period ends, please **do not hesitate to reach out** if you have any additional concerns that you would like us to discuss further.  If you feel that our responses have adequately resolved your concerns, we would be deeply grateful if you might **consider improving your score**.  Your support and recognition are incredibly important to us, and we sincerely thank you for your thoughtful review and constructive input.

---

> ### Author Response · Authors · 2024-11-30
> **Invitation to for the Second Period of Discussion**
>
> Dear Reviewer pafp,
>
> Thank you for your efforts in reviewing and for raising valuable questions. We would like to remind you of the extended discussion period. In the first-round response, we provided detailed replies and a summarized version to **address your concerns** regarding novelty, baseline settings and  provide a **detailed explanation** about Fig. 3.
>
> We sincerely hope you will **reconsider your score** based on these responses, as this is crucial for us, especially since some reviewers have already increased their scores after our clarifications. If you have any further concerns, please do not hesitate to reach out to us, and we will provide additional explanations.

---

> ### Author Response · Authors · 2024-12-01
> **A kind follow up**
>
> Dear Reviewer pafp,
>
> We greatly appreciate your contribution to the review of our manuscript, as well as your valuable suggestions. As the discussion phase is nearing its end, we kindly remind you to comfirm whether our responses have addressed your concerns.
>
> We will deeply appreciate your response and look forward to further discussions to address any remaining concerns you may have.

---

### Official Review · Reviewer_X7hK · 2024-11-02

**Soundness:** 3
**Presentation:** 3
**Contribution:** 3
**Rating:** 5
**Confidence:** 3

**Summary:**

The authors address the challenges of out-of-distribution (OOD) generalization on graphs and propose a model named MPHIL in this paper. MPHIL builds on graph invariant learning (GIL) by disentangling label-correlated invariant subgraphs from environment-specific subgraphs, which helps overcome difficulties in capturing diverse environments and the semantic cliff. For more details, MPHIL first introduces invariant learning within a hyperspherical space using multiple class prototypes to enhance classification without explicit environment modeling. Then, two loss functions ( invariant prototype matching loss and prototype separation loss ) are proposed to ensure accurate class association and inter-class distinction within the hyperspherical space. Extensive experiments conducted across 11 OOD generalization benchmark datasets demonstrate the effectiveness of MPHIL.

**Strengths:**

Different from traditional methods, this paper introduces a novel approach by applying invariant learning in a hyperspherical space to enhance out-of-distribution (OOD) generalization in graph-based tasks. Specially, the invariant prototype matching loss and prototype separation loss proposed in Section 3.4 are well-designed to ensure both accurate class association and inter-class distinction, improving model robustness in hyperspherical space.

**Weaknesses:**

- The novelty of this paper is limited. This paper introduce invariant learning into a hyperspherical space and introduces multiple prototypes to improve class separability, these ideas extend previously established techniques in a relatively straightforward manner. However, the use of hyperspherical embeddings for enhancing model performance of OOD has been explored [1], [2] and [3].

- The experimental results are not convincing, and the baseline methods are not strong enough. Although the authors compare MPHIL with some widely used baseline methods like ERM, CIGA and GSAT, these are neither the latest nor the most competitive, which undermines the significance of the reported performance. More recent OOD methods for graphs should be introduced in the experimental part for fair comparisons, such as OOD-GCL [4], GALA [5] and LECI [6].

[1] Ming Y, Sun Y, Dia O, et al. How to Exploit Hyperspherical Embeddings for Out-of-Distribution Detection?[C]//The Eleventh International Conference on Learning Representations.

[2] Bai H, Ming Y, Katz-Samuels J, et al. HYPO: Hyperspherical Out-Of-Distribution Generalization[C]//The Twelfth International Conference on Learning Representations.

[3] Du X, Gozum G, Ming Y, et al. Siren: Shaping representations for detecting out-of-distribution objects[J]. Advances in Neural Information Processing Systems, 2022, 35: 20434-20449.

[4] Li H, Wang X, Zhang Z, et al. Disentangled Graph Self-supervised Learning for Out-of-Distribution Generalization[C]//Forty-first International Conference on Machine Learning. 2024.

[5] Chen Y, Bian Y, Zhou K, et al. Does invariant graph learning via environment augmentation learn invariance?[J]. Advances in Neural Information Processing Systems, 2024, 36.

[6] Gui S, Liu M, Li X, et al. Joint learning of label and environment causal independence for graph out-of-distribution generalization[J]. Advances in Neural Information Processing Systems, 2024, 36.

**Questions:**

see weakness

---

> ### Author Response · Authors · 2024-11-19
> **Response to Reviewer X7hK (1/2)**
>
> **Q1: The novelty of this paper is limited. This paper introduce invariant learning into a hyperspherical space and introduces multiple prototypes to improve class separability, these ideas extend previously established techniques in a relatively straightforward manner. However, the use of hyperspherical embeddings for enhancing model performance of OOD has been explored [1], [2] and [3].**
>
> **A1:** Thank you for pointing out the application of hyperspherical space in OOD detection [1,3] and OOD generalization [2]. We'd like to emphasize that our work is not a direct extension of previously established techniques (hyperspherical space). Instead of that, we incorporate several **novel designs** driven by **distinct motivations**, underscoring the originality of our approach.
>
> **1. Distinct Motivation.** While the referenced work [2] you mentioned is indeed effective in **image-based** OOD generalization, it rely heavily on the availability of accurate and sufficient environmental information. This critical requirement, however, is **not feasible in the graph domain**, where the environmental information is often unavailable or ambiguous. Our motivation is not merely to extend existing techniques from computer vision to graphs but to address two unique challenges in graph OOD generalization: the inherent difficulty in **capturing environmental information** and **the semantic cliffs across different classes**. These challenges require tailored solutions that go beyond direct application of image-based methods.
>
> Besides, the reference works [1, 3] are focused on OOD detection, which aims to determine whether the test data deviates from the training distribution. In contrast, our work addresses OOD generalization, which is concerned with maintaining robust predictive performance even when the test distribution differs from the training distribution. These are different tasks, and therefore, the challenges to address and the motivations for model design are also different.
>
> **2. Novel Design.**  To address these challenges, we propose a novel method **MPHIL** which contains two key innovations: (1) **hyperspherical invariant learning** for robust feature extraction and prototypical learning in a highly discriminative space. (2)**multi-prototype-based classification** to mitigate the semantic cliff issue and eliminate the need for explicit environment modeling. We propose two novel objective functions: the **invariant prototype matching loss** and the **prototype separation loss** to ensure correct prototype matching and maximize class separability in the hyperspherical space.
>
> We hope our explanation of the differences in motivation and the corresponding solutions above can highlight the unique novelty of our approach and address your concerns. All the references you mentioned we will add to the latest version.

---

> > ### Author Response · Authors · 2024-11-19
> > **Response to Reviewer X7hK (2/2)**
> >
> > **Q2: The experimental results are not convincing, and the baseline methods are not strong enough. Although the authors compare MPHIL with some widely used baseline methods like ERM, CIGA and GSAT, these are neither the latest nor the most competitive, which undermines the significance of the reported performance. More recent OOD methods for graphs should be introduced in the experimental part for fair comparisons, such as OOD-GCL [4], GALA [5] and LECI [6].**
> >
> > **A2:** Thank you for highlighting additional baselines to enhance the comparison and validate MPHIL’s performance. Based on your suggestions, we have added new baselines, including EQuAD [1], LECI [2], and GALA [3] (which were also mentioned in your comments as [2, 3]).  Regarding OOD-GCL (which was mentioned in your comments as [1]), we also considered this method during our baseline selection. Unfortunately, as its code is not publicly available, we regret that we were unable to reproduce its results under our experimental setup for a fair comparison.
> >
> > | Methods| Motif-basis| Motif-size| CMNIST-color| HIV-scaffold|  HIV-size|
> > |   :----:|   :---: |     :---: |   :---: | :---: |:---: |
> > |**MPHIL**| **76.23±4.89** | 58.43±3.15 |41.29±3.85 |**73.94±1.77** |**66.84±1.09** |
> > |EQuAD| 75.46±4.35 | 55.10±2.91 | 40.29±3.95 | 71.49±0.67 | 64.09±1.08 |
> > |LECI| 73.65±2.21 | **62.18±1.84** | **42.88±2.61** | 72.01±0.91 | 63.12±1.62 |
> > |GALA| 72.97±4.28 | 60.82±0.51| 40.62±2.11| 71.22±1.93| 65.29±0.72|
> >
> > | Methods| Ic50-assay| Ic50-scaffold| Ic50-size|Ec50-assay| Ec50-scaffold| Ec50-size
> > |   :----:|   :---: |     :---: |   :---: | :---: |:---: |:---: |
> > |**MPHIL**| **72.96±1.21** | **68.62±0.78** |**68.06±0.55** |**78.08±0.54** |**68.34±0.61** | **68.11±0.58** |
> > |EQuAD| 71.57±0.95 | 67.74±0.57 | 67.54±0.27 | 77.64±0.63 | 65.73±0.17 | 64.39±0.67 |
> > |LECI| 71.57±0.92 | 65.01±0.78 | 64.79±0.86 | 73.51±0.62 | 62.89±0.82 | 63.47±0.95 |
> > |GALA| 70.58±2.63 | 66.35±0.86| 66.54±0.93| 77.24±2.17| 66.98±0.84|63.71±1.17 |
> >
> > The results shows that **MPHIL still achieves the best average performance**. In our experiments, we found that LECI does outperform MPHIL on certain datasets, but this is because LECI **relies on true environment labels**. Specifically, when the benchmark provides carefully designed graph environment partitions, LECI can effectively leverage this information for OOD generalization. However, in real-world applications, this is often not feasible. Therefore, MPHIL **does not rely on any environment-related information** from the benchmark but instead leverages **hyperspherical invariant feature learning** and the **multi-prototype classification mechanism** to ensure the learning of environment-independent and maximally class-separable invariant features, thereby achieving graph OOD generalization.
> >
> > [1] Yao T, Chen Y, Chen Z, et al. Empowering graph invariance learning with deep spurious infomax[J]. arXiv preprint arXiv:2407.11083, 2024.
> >
> > [2] Gui S, Liu M, Li X, et al. Joint learning of label and environment causal independence for graph out-of-distribution generalization[J]. Advances in Neural Information Processing Systems, 2024, 36.
> >
> > [3] Chen Y, Bian Y, Zhou K, et al. Does invariant graph learning via environment augmentation learn invariance?[J]. Advances in Neural Information Processing Systems, 2024, 36.

---

> ### Author Response · Authors · 2024-11-22
> **Invitation to Discussion**
>
> Dear Reviewer X7hK:
>
> We sincerely appreciate your valuable contributions during the review process and your recognition of and suggestions for our work. To save you time, we provide a **summarized version** of our response to facilitate a quicker understanding:
>
> - **Emphasis on Novelty**: Regarding the novelty of our approach, we emphasize its unique motivation and design, distinct from existing hypersphere learning methods. Unlike OOD generalization in computer vision (e.g., the reference [2] you mentioned), graph data presents unique challenges, including **limited access to environmental information** and separability issues caused by the **semantic cliff**. To address these, we introduce two key modules—**hyperspherical invariant learning** and **multi-prototype-based classification**—alongside two novel loss functions. These innovations enable **intra-class consistency** and **inter-class separability**, supporting **maximally class-separable, environment-free graph OOD generalization**.
>
> - **Supplemental Baseline Models**: We added three state-of-the-art graph OOD generalization methods for comparison, as per your suggestion. The experimental results confirm that MPHIL **continues to achieve superior performance**.
>
> Finally, we have included the references you pointed out, recognizing their value in enhancing our work.
>
> We have carefully **response all your questions** and truly appreciate the opportunity to clarify and improve our work based on your feedback.  With some time remaining before the rebuttal period ends, please **do not hesitate to reach out** if you have any additional concerns that you would like us to discuss further.  If you feel that our responses have adequately resolved your concerns, we would be deeply grateful if you might **consider improving your score**.  Your support and recognition are incredibly important to us, and we sincerely thank you for your thoughtful review and constructive input.

---

> ### Author Response · Authors · 2024-11-30
> **Invitation to for the Second Period of Discussion**
>
> Dear Reviewer X7hK,
>
> Thank you for your efforts in reviewing and for raising valuable questions. We would like to remind you of the extended discussion period. In the first-round response, we provided detailed replies and a summarized version to **address your concerns** regarding novelty and baseline settings.
>
> We sincerely hope you will **reconsider your score** based on these responses, as this is crucial for us, especially since some reviewers have already increased their scores after our clarifications. If you have any further concerns, please do not hesitate to reach out to us, and we will  provide additional explanations.

---

### Official Review · Reviewer_gRW6 · 2024-11-04

**Soundness:** 2
**Presentation:** 2
**Contribution:** 2
**Rating:** 5
**Confidence:** 4

**Summary:**

The paper presents a novel method MPHIL to address two challenges in existing graph invariant learning methods: the difficulty of capturing environmental information and the semantic cliff  across different classes. MPHIL operates in hyperspherical space with class prototype as intermediate variable to improve representation separability and robustnes without explicit environment modeling.
Two innovative loss functions are derives to enhance intra-class invariance and inter-class separability. Extensive experiments show that MPHIL outperforms state-of-the-art methods on various benchmark datasets, demonstrating its effectiveness for OOD generalization.

**Strengths:**

This paper propose a new method for graph invariant learning.
- The paper is well-organized.

**Weaknesses:**

-  Intuitively, the accuracy the class prediction relies on the fastest prototype point in the same class prototype set.  What if use the mean/sum/min for predicting the class? More ablation study can make your work more convincing.
- The novelty of this proposed method is limited. I don't think the projection to the hypersphere here is novel. Actually, it is a simple cosine distance. How about using other distances like L2, L1, more analysis should be included.
- I wonder the performance when only using one protype.
- The prototype updating mechanism is complicated, what if directly use the representation of the sample from training set after training.
- The explaination of some symbols is missing, e.g. the $\omega_k^{(c)}$ in Eq (12).

**Questions:**

See weakness.

---

> ### Author Response · Authors · 2024-11-19
> **Response to Reviewer gRW6 (1/2)**
>
> **Q1: Intuitively, the accuracy the class prediction relies on the fastest prototype point in the same class prototype set. What if use the mean/sum/min for predicting the class? More ablation study can make your work more convincing.**
>
> **A1:** Thanks for your valuable comments! As you mentioned, intuitively, the prototype most similar to the sample provides the most representative and discriminative information, facilitating faster convergence of the sample towards its correct class space. To further justify this intuition, we used to conducted experiments replacing the **max** operation with **min**, **median**, **sum**, and **mean** in our approach （Equation 12）.
>
> | metrics| GOODHIV-Scaffold | DrugOOD-IC50-Size| CMNIST-Size
> |   :----:|   :---: |     :---: |   :---: |
> |**max**| **73.94±1.77** | **68.06±0.55** |**41.29±3.85** |
> |median| 71.39±1.75 | 65.64±0.61 | 32.57±4.06 |
> |sum| 68.37±1.65 | 58.61±0.44 | 31.88±3.74 |
> |mean| 70.35±1.50 | 63.53±0.48| 28.51±5.57 |
> |min| 66.76±1.82 | 64.58±0.78 | 21.97±6.91 |
>
> The results show that the **max** operation consistently achieves the best performance, followed by **median** with **min** performing the worst. We believe this is because the **median** is relatively robust to outliers and can still capture central tendencies to some extent, while **sum** and **mean** dilute the impact of the most relevant prototype by averaging contributions from all prototypes. The **min** operation, on the other hand, focuses on the least similar prototype, which misrepresents the true relationship between the sample and its class and slows down the convergence of the optimization process.
>
> **Q2: The novelty of this proposed method is limited. I don't think the projection to the hypersphere here is novel. Actually, it is a simple cosine distance. How about using other distances like L2, L1, more analysis should be included.**
>
> **A2:** Thanks for your valuable comments! While hyperspherical learning may initially appear similar to cosine similarity, its core principle is the normalization of high-dimensional representations into unit vectors on a hypersphere. This approach suppresses amplitude interference and optimizes angular relationships, thereby **enhancing intra-class consistency** and **maximizing inter-class separability**. These properties make hyperspherical learning, though seemingly simple, particularly effective for OOD generalization and has been demonstrated in computer vision, while our ablation experiments (**w/o project**) also illustrate its effectiveness for graph OOD methods.
>
>
> While a direct application of hyperspherical space may have limited novelty, we want to emphasize that the primary motivation of our paper is not a simple application of hyperspherical spaces but rather to tackles two unique challenges in graph OOD generalization: **the difficulty in capturing environmental information** and **the semantic cliff across different classes.** To overcome these challenges, we further propose two novel and effective designs, i.e., **hyperspherical invariant feature learning** and **multi-prototype classification mechanism**. These components work together to ensure the learning of environment-independent, maximally class-separable invariant features. Ablation studies (**w/o Inv.Enc.**, **w/o Multi-P**) demonstrate their critical contributions, showcasing how our method advances the state-of-the-art in graph OOD generalization.
>
> | metrics| GOODHIV-Scaffold | DrugOOD-IC50-Size| CMNIST-Size
> |   :----:|   :---: |     :---: |   :---: |
> |**MPHIL**| **73.94±1.77** | **68.06±0.55** |**41.29±3.85** |
> |w/o project | 65.78±3.57 | 51.96±2.54 | 21.05±4.89 |
> |w/o Inv.Enc| 66.72±1.19 | 63.73±0.89 | 34.86±2.92 |
> |w/o Multi-P| 62.11±1.95 | 57.64±1.02| 20.58±3.78 |
>
> Furthermore, MPHIL does involve multiple similarity computations (Eq. 12-14), all based on cosine similarity. Following your suggestion, we further replaced cosine similarity with $L1$, $L2$, and $L_\infty$ distances to analyze the impact of different similarity metrics. The results are summarized in the table below.
>
> | metrics| GOODHIV-Scaffold | DrugOOD-IC50-Size| CMNIST-Size
> |   :----:|   :---: |     :---: |   :---: |
> |**cosine**| **73.94±1.77** | **68.06±0.55** |**41.29±3.85** |
> |L1| 63.66±1.92 | 55.91±0.56 | 30.18±3.77 |
> |L2| 69.66±1.74 | 57.60±0.48 | 32.94±3.26 |
> |L$\infty$| 60.05±1.29 | 58.79±0.81| 38.43±4.11 |
>
> The experimental results demonstrate that cosine similarity is the most effective measure, as it leverages the directional properties of hyperspherical space to distinguish features with high precision. In contrast, measures like $L1$, $L2$, and $L_\infty$ distances primarily focus on magnitude differences, which are less relevant in hyperspherical representations where all features are normalized to have unit norms.

---

> ### Author Response · Authors · 2024-11-19
> **Response to Reviewer gRW6 (2/2)**
>
> **Q3: I wonder the performance when only using one prototype.**
>
> **A3**: Thanks for your valuable comments! In the ablation study of the original manuscript (Table 2), we included the results for 'w/o Multi-P,' which corresponds to using only a single prototype for classification. The results indicate that this approach significantly impacts model performance, as it blurs decision boundaries and prevents the loss function $\mathcal{L}_{\mathrm{PS}}$ from being effectively optimized, ultimately compromising inter-class separability.
> | metrics| GOODHIV-Scaffold | DrugOOD-IC50-Size| CMNIST-Size
> |   :----:|   :---: |     :---: |   :---: |
> |**MPHIL**| **73.94±1.77** | **68.06±0.55** |**41.29±3.85** |
> |w/o Multi-P| 62.11±1.95 | 57.64±1.02 |20.58±3.78 |
>
> **Q4: The prototype updating mechanism is complicated, what if directly use the representation of the sample froraining set after training.**
>
> **A4:** Thanks for your valuable comments! In the original manuscript, we conducted an ablation study (Table 2, w/o Update) to investigate the impact of removing the prototype update mechanism described in Section 3.3. Specifically, this is equivalent to directly using the mean of same-class samples as the prototype (the number of prototypes for each class is reduced to one). To further address your concerns, we additionally designed two alternative prototype construction mechanisms:
> (1) Randomly selecting $K$ samples from each class as prototypes.
> (2) Clustering the samples of each class and using the $K$ cluster centers as prototypes.
> The experimental results are shown in the table below.
> | methods| GOODHIV-Scaffold | DrugOOD-IC50-Size| CMNIST-Size
> |   :----:|   :---: |     :---: |   :---: |
> |**MPHIL**| **73.94±1.77** | **68.06±0.55** |**41.29±3.85** |
> |mean| 67.89±1.84 | 64.14±1.22 | 38.95±3.01 |
> |random| 57.64±1.23 | 52.12±0.69 | 25.52±4.59 |
> |kmeans| 68.11±1.92 | 60.82±0.51| 40.12±3.91 |
>
> The results show that the random sampling method leads to the most significant performance drop, as it fails to effectively ensure inter-class separability. While the mean and k-means methods perform slightly better, they still struggle to generate optimal prototypes due to their inability to fully capture the intra-class diversity. This further demonstrates the superiority of our proposed prototype update method.
>
> **Q5: The explaination of some symbols is missing, e.g. the $w_{k}^{c}$ in Eq (12).**
>
> **A5:** Thanks for your valuable comments, and we apologize for the oversight in explaining some of the notation. We have now clarified this in the latest manuscript (page 7 line 324). Specifically, $w_{k}^{c}$ denotes the weight assigned to the $k$-th prototype of class $c$ for the current sample.

---

> ### Author Response · Authors · 2024-11-22
> **Invitation to Discussion**
>
> Dear Reviewer gRW6:
>
> We sincerely appreciate your valuable contributions during the review process and your recognition of and suggestions for our work. To save you time, we provide a **summarized version** of our response to facilitate a quicker understanding:
>
> - **Expanded Ablation Experiments**: As per your suggestion, we replaced the max operation with other statistics, substituted cosine similarity with alternative similarity measures, and explored additional prototype update strategies. The results of these experiments **consistently validate the effectiveness of our proposed design**.
> - **Clarification on Novelty**: While you expressed concerns about the simplicity and novelty of hyperspherical space, we further explain its unique properties and why it is particularly well-suited for OOD generalization tasks. More importantly, our proposed MPHIL **is far from a simple application of hyperspherical spaces**. It directly tackles two critical challenges in graph OOD generalization: **the difficulty in capturing environmental information** and the **semantic cliff across different classes**. Our contributions include **hyperspherical invariant feature learning** and a **multi-prototype classification mechanism**, supported by **two novel loss functions**. These innovations ensure **intra-class consistency** and **inter-class separability**, enabling us to achieve better performance in graph OOD generalization.
>
> We have carefully **response all your questions** and truly appreciate the opportunity to clarify and improve our work based on your feedback.  With some time remaining before the rebuttal period ends, please **do not hesitate to reach out** if you have any additional concerns that you would like us to discuss further.  If you feel that our responses have adequately resolved your concerns, we would be deeply grateful if you might **consider improving your score**.  Your support and recognition are incredibly important to us, and we sincerely thank you for your thoughtful review and constructive input.

---

> > ### Comment · Reviewer_gRW6 · 2024-11-22
> >
> > Thanks for the rebuttal. I am still concerned about the novelty of the proposed method.  I keep my score.

---

> > > ### Author Response · Authors · 2024-11-22
> > > **Response to Reviewer gRW6**
> > >
> > > Thank you for your response. We understand that your primary concern about novelty is that you **"don't think the projection to the hypersphere here is novel"**. However, this is not the core contribution of our approach. The key innovation of MPHIL lies in enabling out-of-distribution generalization for graphs with **maximum class separability**, **even in the absence of environmental information**—a capability that existing methods **cannot achieve**. This represents the most significant contribution of our work.
> > >
> > > If you have further questions or require additional clarification regarding this aspect of novelty, please do not hesitate to reach out. Thank you once again for your valuable feedback, and we look forward to hearing from you.

---

> ### Author Response · Authors · 2024-11-30
> **Invitation to for the Second Period of Discussion**
>
> Dear Reviewer gRW6
>
> Thank you for your efforts in reviewing and for raising valuable questions. We would like to remind you of the extended discussion period. In the first-round response, we provided detailed replies and a summarized version to address your concerns regarding novelty and requests for more ablation experiments.
>
> Thank you for your previous response, but we would still **like to know your specific concerns about novelty**, or **why our previous answer did not address your concerns**. We hope to **do our best** to solve your confusion to prompt you to **reconsider the score**, as this is crucial for us, especially since some reviewers have already increased their scores after our clarifications. If you have any further concerns, please do not hesitate to reach out to us, and we will provide additional explanations.

---

### Author Response · Authors · 2024-11-19
**Response  Summary**

We sincerely thank all the reviewers for their valuable and insightful comments. We are delighted that the reviewers acknowledged the importance of the research problem addressed in this paper (Reviewer pafp), recognized the originality of our proposed method (Reviewers s53C and X7hK) and the extensive experiments conducted (Reviewers pafp and s53C), and found the paper to be logically clear and well-organized (Reviewers gRW6 and s53C).

To the best of our efforts, we have provided detailed responses below to address each of the reviewers' concerns. Additionally, we have carefully revised the manuscript based on their feedback and highlighted the changes in blue.

Specifically, the main responses and modifications we made are as follows.
-  We have further emphasized the **unique motivation and contributions** of the proposed MPHIL framework to demonstrate its sufficient novelty.
-  We incorporated **three state-of-the-art baseline methods** and updated the experimental results, showing that MPHIL consistently achieves the best average performance.
-  We have further explained the Fig. 3 and included **additional quantitative analyses** to better illustrate the advantages of our method.
-  We conducted **more detailed ablation studies** to validate the effectiveness of MPHIL's design and provided further clarification on the principles of multi-prototype classification.

---

### Meta-Review · Area_Chair_w9zD · 2024-12-22

**Metareview:**

This paper proposes a method for graph out-of-distribution generalization using hyperspherical invariant learning and multi-prototype classification, supported by experiments across various benchmarks. The paper addresses a meaningful problem and is well-structured. The reviewers raised concerns about the method's novelty and the competitiveness of its baseline comparisons. Despite the authors’ thorough rebuttal and additional clarifications, three reviewers rated the paper below the acceptance threshold. Respecting the majority opinion, I lean towards rejection.

**Additional Comments On Reviewer Discussion:**

The discussion phase highlighted several key concerns raised by reviewers, including questions about the novelty of the proposed methodology, the clarity of certain visualizations, and the inclusion of additional baselines for comparison. The authors provided detailed responses, emphasizing the distinct motivations and innovations of their approach, conducting additional ablation studies, and including comparisons with recent state-of-the-art methods. They also revised the manuscript to address formula clarity, improved figure explanations, and updated the code repository for reproducibility.

---

### Decision · Program_Chairs · 2025-01-22

Reject